

# Dynamically restoring conformal invariance in (integrable) $\sigma$-models

Rigers Aliaj[1★], Konstantinos Sfetsos[1,2†] and Konstantinos Siampos[1‡]

**1** Department of Nuclear and Particle Physics, Faculty of Physics,
National and Kapodistrian University of Athens, Athens 15784, Greece
**2** Theoretical Physics Department, CERN, 1211 Geneva 23, Switzerland

★ raliaj@phys.uoa.gr , † ksfetsos@phys.uoa.gr , ‡ konstantinos.siampos@phys.uoa.gr

## Abstract

Integrable $\lambda$-deformed $\sigma$-models are characterized by an underlying current algebra/coset model CFT deformed, at the infinitesimal level, by current/parafermion bilinears. We promote the deformation parameters to dynamical functions of time introduced as an extra coordinate. It is conceivable that by appropriately constraining them, the beta-functions vanish and consequently the $\sigma$-model stays conformal. Remarkably, we explicitly materialize this scenario in several cases having a single and even multiple deformation parameters. These generically obey a system of non-linear second-order ordinary differential equations. They are solved by the fixed points of the RG flow of the original $\sigma$-model. Moreover, by appropriately choosing initial conditions we may even interpolate between the RG fixed points as the time varies from the far past to the far future. Finally, we present an extension of our analysis to the Yang–Baxter deformed PCMs.

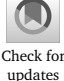

# 1 Introduction

Two-dimensional conformal $\sigma$-models play an important rôle in string theory, since they provide consistent backgrounds for propagation of strings in curved spacetimes. As usual these models are described in terms of the action[1]

$$S = \frac{1}{2\pi} \int d^2\sigma (G_{MN} + B_{MN}) \partial_+ X^M \partial_- X^N, \quad M = 1, 2, \ldots, d, \tag{1}$$

from which we read off the background fields for the metric $G_{MN}$ and the antisymmetric tensor $B_{MN}$. In addition, there is also a dilaton field $\Phi$. Demanding conformal invariance at one-loop order leads to vanishing beta-functions for the metric and the antisymmetric fields. At one-loop these read [1–7]

$$R_{MN} - \frac{1}{4} H_{MKL} H_N{}^{KL} + 2\nabla_M \partial_N \Phi = 0, \quad \nabla^P(e^{-2\Phi} H_{MNP}) = 0, \tag{2}$$

where $R_{MN}$ is the Ricci tensor and $\nabla_M$ is the covariant derivative built out of the metric $G_{MN}$, using the Levi–Civita connection. In addition, $H_{MNP} = \partial_M B_{NP} + \text{cyclic}$ is the field strength of the antisymmetric tensor $B_{MN}$. The dilaton beta-function requires that the expression

$$R - \frac{1}{12} H_{MNP} H^{MNP} + 4\nabla^2 \Phi - 4(\partial \Phi)^2 = w, \tag{3}$$

when (2) is applied, is a constant labelled here by $w$. Finally, we may compute the central charge at one-loop order from the dilaton beta-function or the Weyl anomaly coefficient (3) as [8,9]

$$W = d - 3w. \tag{4}$$

---

[1]The world-sheet coordinates $\sigma^\pm$ and $(\tau, \sigma)$ are given by

$$\sigma^\pm = \tau \pm \sigma, \quad \partial_\pm = \frac{1}{2}(\partial_\tau \pm \partial_\sigma), \quad d^2\sigma = d\tau \, d\sigma.$$

In this work we will consider the (multi) $\lambda$ deformed current algebra CFTs [10], the $\lambda$-deformed coset CFTs [10, 11] as well as, to a lesser extend, the $\eta$-deformed PCMs [12], where we will promote the deformation parameters to dynamical functions of time[2]. We will demand that these functions are constrained in such a way that the resulting $\sigma$-model stays conformal at one-loop order (2). This is not a priori a consistent procedure warranted to lead to a conformal model, especially for a multi-parameter deformation. However, we will present several cases in which this yields a consistent system of non-linear second-order ordinary differential equations. These equations are trivially solved by the fixed points of the RG flows of the original $\sigma$-model, but, more interestingly, they also admit time dependent solutions interpolating between these fixed points. Let us point out that there is no direct identification of the RG scale with the target-space time as the RG equations do not solve the system of second-order ordinary differential equations. Yet the system of second-order ordinary differential equations upon demanding an appropriate behaviour may interpolate between the fixed points. A similar set-up was considered in [13–15] with the crucial difference that in these works the starting point was a CFT and the demand was that the dynamical promotion led again to a conformal model at one-loop order.

This work is structured in two classes of models, (I) and (II):

In models of class (I), considered in Sections 2, 3 and 4 we study integrable $\sigma$-models known as $\lambda$-deformations [10], which interpolate between a (gauged) WZW model and the non-Abelian T-dual of a PCM. These models are dynamically extended by adding an extra coordinate $t$ contributing to its Lagrangian density, letting $\lambda$ depending on $t$ and including a time dependent term in the dilaton. More specifically, in Section 2, we considered the $\lambda$-deformed $SU(2)_k/U(1)$ [10]. In Section 3, we worked out the scale invariant deformation of the $SL(2,\mathbb{R})_{-k}/SO(2)$ coset CFT [16]. The resulting system of differential equations can be analytically integrated. In Section 4, we studied the $\lambda$-deformed $SU(2)_k$ case for an isotropic and a marginal deformation [10]. In the latter case we make contact with the Nappi–Witten expanding universe $\frac{SU(2)_k \times SL(2,\mathbb{R})_{-k}}{U(1) \times U(1)}$ [17]. Lastly, in Section 7 we extend our analysis to the Yang–Baxter deformed $SU(2)$ PCM [12].

In models of class (II), considered in Sections 5 and 6 we dynamically extend (integrable) $\sigma$-models which interpolate between exact (coset) CFTs. In Section 5, we considered the single $\lambda$-deformed $G_{k_1} \times G_{k_2}$ for an isotropic, a marginal and a generic deformation $\lambda_{ab}$ [18]. In Section 6, we studied the $\lambda$-deformed $SU(2)_{k_1} \times SU(2)_{k_2}/SU(2)_{k_1+k_2}$ [19]. Finally, in Appendix A we gathered the technical details of the generic deformation $\lambda_{ab}(t)$ considered in Section 5.

## 2  $\lambda$-deformed $SU(2)_k/U(1)$

In this Section we consider the simplest example, namely that for the $\lambda$-deformed $SU(2)_k/U(1)$ [10] whose action is given by

$$
\begin{aligned}
S = \frac{k}{\pi} \int \mathrm{d}^2\sigma \Big\{ & \frac{1-\lambda}{1+\lambda} \big( \partial_+\beta \partial_-\beta + \cot^2\beta \, \partial_+\alpha \partial_-\alpha \big) \\
& + \frac{4\lambda}{1-\lambda^2} \big( \sin\alpha \cot\beta \, \partial_+\alpha + \cos\alpha \, \partial_+\beta \big) \big( \sin\alpha \cot\beta \, \partial_-\alpha + \cos\alpha \, \partial_-\beta \big) \Big\} .
\end{aligned}
\tag{5}
$$

Including a diffeomorphism induced by the scalar

$$
\Phi = -\ln \sin\beta ,
\tag{6}
$$

---

[2]In practice we promote the target space metric $G_{\mu\nu}$ with coupling constants $\lambda^i$ to time-dependent ones

$$
G_{\mu\nu}(X^\mu; \lambda^i)\mathrm{d}X^\mu \mathrm{d}X^\nu \quad \Longrightarrow \quad \mathrm{d}t^2 + G_{\mu\nu}(X^\mu; \lambda^i)\mathrm{d}X^\mu \mathrm{d}X^\nu ,
$$

similarly for the antisymmetric tensor $B_{\mu\nu}$ and the dilaton $\Phi$.

the above $\sigma$-model is renormalizable at one-loop in the $1/k$ expansion and its RG flow is given by [20, 21]

$$\frac{\mathrm{d}\lambda}{\mathrm{d}\ln\mu^2} = -\frac{\lambda}{k}, \tag{7}$$

where $\mu$ is the energy scale. From the above expression we find that the operator $\mathcal{O}$, driving the perturbation, is relevant and has scaling dimension equal to $\Delta_{\mathcal{O}} = 2(1 - 1/k)$. This is in accordance with general considerations for the case at hand

$$\frac{\mathrm{d}\lambda}{\mathrm{d}\ln\mu^2} = \frac{\Delta_{\mathcal{O}}-2}{2}\lambda + \frac{1}{2}C_{\mathcal{O}\mathcal{O}}{}^{\mathcal{O}}\lambda^2 + \mathcal{O}(\lambda^3), \tag{8}$$

where $C_{\mathcal{O}\mathcal{O}}{}^{\mathcal{O}}$ denotes the OPE coefficient of the operator $\mathcal{O}$ with itself. In our case this operator is the parafermion [22] bilinear corresponding to the coset CFT. To leading order in the $1/k$-expansion there are no higher in $\lambda$-corrections since $C_{\mathcal{O}\mathcal{O}}{}^{\mathcal{O}}$ vanishes due to the fact that the coset CFT corresponds to a symmetric space.

## 2.1 The Euclidean case

Let us now dynamically promote the above model by adding an extra spacelike coordinate contributing to the Lagrangian the term $k/\pi\,\partial_+ t\partial_- t$ and moreover allowing the constant $\lambda$ to depend on $t$. In addition, we add to the scalar (6) the term $\Phi_0(t)$ and consider the entire $\Phi$ as the dilaton of the resulting $\sigma$-model. Then by imposing the one-loop beta-function equations for the metric (2) we find the following differential equations for $\lambda(t)$ and $h(t) = \dot{\Phi}_0(t)$

$$\ddot{\lambda} = -4\lambda + 2\dot{\lambda}\left(h - \frac{\lambda\dot{\lambda}}{1-\lambda^2}\right), \quad \dot{h} = \frac{\dot{\lambda}^2}{(1-\lambda^2)^2}. \tag{9}$$

In addition, the dilaton beta-function (3) yields the constant

$$-\frac{1}{k}\left(2h^2 - \dot{h}\right) + \frac{2}{k}\frac{1+\lambda^2}{1-\lambda^2} = w, \tag{10}$$

which is nothing but a first integral of (9). This system, as well as the corresponding action, is invariant under $t \to -t$ and in addition, they are invariant under the two symmetries

$$\begin{aligned} \text{I:} \quad & \lambda \to \lambda^{-1}, \quad k \to -k, \quad t \to it, \\ \text{II:} \quad & \lambda \to -\lambda, \quad \alpha \to \alpha \pm \frac{\pi}{2}. \end{aligned} \tag{11}$$

This is the analogue of the symmetries of the action (5) and of the beta-function (7) found in [20, 23].

Note that (9) has the trivial solution with $\lambda(t) = 0, h(t) = $ constant, corresponding to the $SU(2)_k/U(1) \times \mathbb{R}_Q$ CFT. It is instructive to see how at the linear level, for small $\lambda$, the perturbation stays marginal. In the integrable case in which $\lambda$ is constant the CFT perturbation is bilinear in the compact parafermions of the coset theory and has dimension equal to $\Delta_{\mathcal{O}} = 2(1 - 1/k)$, i.e. a relevant operator. This operator is multiplied by $\lambda$ which does not of course affect this. However, in the dynamical case $\lambda(t)$ acquires a dimension itself. Indeed, the first of (9) can be easily seen to be approximated as $t \to -\infty$ by the equation corresponding to a harmonic oscillator with friction. It is solved approximately by (recall that $h = \dot{\Phi}_0$)

$$\lambda(t) \simeq c\,e^{a_- t}, \quad \Phi_0(t) \simeq h_i t, \quad a_- = h_i - \sqrt{h_i^2 - 4} > 0, \tag{12}$$

where $c$ and $h_i$ are integration constants and the corrections are $\mathcal{O}\left(e^{2a_-t}\right)$. Reality of the dilaton and demanding a weak string coupling, i.e. $e^{\Phi(t)} \ll 1$, as $t \to -\infty$, gives the condition $h_i > 2$. We may also obtain an approximate solution for $0 < h_i < 2$

$$\lambda(t) \simeq c\, e^{h_i t} \sin\left[\sqrt{4-h_i^2}\,(t-t_0)\right], \quad \Phi_0(t) \simeq h_i t\,, \tag{13}$$

again with correction of $\mathcal{O}\left(e^{2h_i t}\right)$. For the critical value of the "friction" coefficient $h_i = 2$ the solution is

$$\lambda(t) \simeq e^{2t}\left(c + \tilde{c}\,t\right), \quad \Phi_0(t) \simeq 2t\,, \tag{14}$$

with $\mathcal{O}\left(e^{4t}\right)$ corrections.

To read the scaling dimension of $\lambda(t)$ we first pass to Euclidean signature

$$\tau \to -i\tau\,, \quad \sigma^+ \to -iz\,, \quad \sigma^- \to -i\bar{z}\,, \quad \partial_+ \to i\partial\,, \quad \partial_- \to i\bar{\partial}\,, \quad d^2z = d\tau\, d\sigma\,, \tag{15}$$

where $z = \tau + i\sigma, \bar{z} = \tau - i\sigma$, with the action in the path integral appearing as $e^{-S}$. The scaling dimension of $\lambda(t)$ can be read throughout the general formulae for an exponential operator. Indeed for the linear dilaton theory (flat worldsheet)

$$S_{\ell.d.} = \frac{s}{\pi\alpha'}\int d^2z\, \partial X \bar{\partial}X\,, \quad \Phi_0 = QX\,, \tag{16}$$

where $s = \pm 1$, parameterizing a spacelike or a timelike boson $X$, respectively, we have the energy–momentum tensor

$$T_{zz} = -\frac{s}{\alpha'}(\partial X)^2 + Q\,\partial^2 X\,, \quad T_{z\bar{z}} = -Q\,\partial\bar{\partial}X\,, \tag{17}$$

with central charge given by $c_{\ell.d.} = 1 + 6s\alpha'Q^2$. Then, for an exponential operator we have that

$$V_{\Delta,\bar{\Delta}} =: e^{aX} :\,, \quad \Delta = \bar{\Delta} = \frac{s a \alpha'}{2}\left(Q - \frac{a}{2}\right)\,, \tag{18}$$

where $\Delta$ and $\bar{\Delta}$ are the holomorphic and the antiholomorphic dimensions, respectively. In the case at hand, we have a background charge dilaton.

Next we read off various parameters either from (12), (13) or from (14) depending on whether or not the positive constant $h_i$ is larger, smaller or equal to 2. We will give the expressions corresponding to the first case. The results for the others are identical. We obtain that

$$X = t\,, \quad a = a_-\,, \quad s = 1\,, \quad \alpha' = \frac{1}{k}\,, \quad Q = h_i\,. \tag{19}$$

Hence, the holomorphic and antiholomorphic dimensions $(\Delta, \bar{\Delta})$ equal to $1/k$. When these dimensions and the scaling dimension of the parafermion bilinear $\Delta_{\mathcal{O}} = 2(1 - 1/k)$ are added up we find the scaling dimension of a marginal operator. Hence, at $t \to -\infty$ we find the $SU(2)_k/U(1) \times \mathbb{R}_Q$ CFT. This property is supported by the Weyl anomaly constant coefficient (4) for the case at hand (10). Using (12) equals to

$$\text{computed as } t \to -\infty: \quad W = 3 - 3w = 3 - \frac{6}{k} + \frac{6h_i^2}{k} = c_{2d} + c_{\ell.d.}\,, \tag{20}$$

where we have decomposed the latter expression in the two contributions, namely that of the coset CFT $SU(2)_k/U(1)$

$$c_{2d} = \frac{3k}{k+2} - 1 = 2 - \frac{6}{k} + \mathcal{O}\left(\frac{1}{k^2}\right)\,, \tag{21}$$

and that of the linear dilaton CFT at charge $h_i$. Note that, preserving conformal invariance at linear order does not warrant its preservation to all orders, that is consistency conditions of the form (9) are not always possible to be obtained. In that sense, this as well as the rest of the examples presented in the present paper are quite intriguing.

An interesting question is what happens as time progresses starting from the remote past. We expect that the model will approach the strong coupling regime in which $\lambda(t)$ approaches unit. This is analogous to the behaviour of the model with constant $\lambda$ under RG flow which in the IR reaches $\lambda = 1$, which is the strong coupling region. Letting $\lambda(t) = 1 - \alpha(t)$ we find that for small $\alpha(t)$ the system (9) becomes

$$\ddot{\alpha} \simeq 2(2 + h\dot{\alpha}) + \frac{\dot{\alpha}^2}{\alpha}, \quad \dot{h} \simeq \frac{\dot{\alpha}^2}{4\alpha^2}. \tag{22}$$

In addition the dilaton equation gives that

$$\dot{h} - 2h^2 + \frac{2}{\alpha} \simeq kw. \tag{23}$$

We may easily verify that for small $t$ we have the approximate solution

$$t \to 0^- : \quad \alpha \simeq 2t^2, \quad \Phi_0 \simeq -\ln(-t), \tag{24}$$

which corresponds to the constant $w = 0$ in (10). Since the latter is the same as that in (21) we find that the constant appearing in the approximate solution valid in the remote past $h_i = 1$. Hence, the string coupling behaves as $e^{\Phi(t)} \sim -1/t$, as $t \to 0^-$ (obviously, this value for $t$ can be shifted at will) and the model reaches the strong coupling region. In that regime one may resort to the non-Abelian T-dual limit, in which $\lambda(t) \to 1$ and the overall level coefficient $k \to \infty$ in a correlated manner, so that the model still makes sense. The limiting procedure here is identical to the constant $\lambda$ case [10], it can easily be performed and therefore will not repeat it here. We have numerically checked that the $\lambda(t) \to 1^-$ limit is reached monotonically from the far past at $t \to -\infty$ towards $t = 0$.

## 2.2 The Lorentzian case

In this case we add to the Lagrangian density (1) the term $-k/\pi \, \partial_+ t \partial_- t$. Hence, our results should be obtained from the above Euclidean case by the analytic continuation $t \to it$ (under this $h \to -ih$). Then from (9) and (10) we obtain the system of equations given by

$$\ddot{\lambda} = 4\lambda + 2\dot{\lambda}\left(h - \frac{\lambda\dot{\lambda}}{1 - \lambda^2}\right), \quad \dot{h} = \frac{\dot{\lambda}^2}{(1 - \lambda^2)^2}. \tag{25}$$

As before, this has the trivial solution $\lambda(t) = h(t) = 0$, corresponding to the $SU(2)_k/U(1)$ coset CFT and the free scalar $t$ at zero background charge. In addition, the dilaton beta-function requires that (3) is a constant. Explicitly,

$$\frac{1}{k}\left(2h^2 - \dot{h}\right) + \frac{2}{k}\frac{1 + \lambda^2}{1 - \lambda^2} = w. \tag{26}$$

Similarly, to the Euclidean case we can solve (25) approximately as $t \to -\infty$

$$\lambda(t) \simeq c\, e^{a_+ t}, \quad \Phi_0(t) \simeq h_i t, \quad a_+ = h_i + \sqrt{h_i^2 + 4} > 0, \tag{27}$$

where $c$ and $h_i > 0$ are integration constants and corrections of $\mathcal{O}(e^{a_+ t})$. Hence, the model is at the weak coupling regime, i.e. $e^{\Phi(t)} \ll 1$. In addition, following the analysis of the Euclidean

case the holomorphic and anti-holomorphic dimension of the solution $\lambda(t)$ can be read using (15), (16) and (18), where in the case at hand

$$X = t, \qquad a = a_+, \qquad s = -1, \qquad \alpha' = \frac{1}{k}, \qquad Q = h_i, \tag{28}$$

leading to holomorphic and antiholomorphic dimensions $(\Delta, \bar{\Delta})$ equal to $1/k$. When these are added up they precisely provide the central charge deficit $2/k$ and the perturbation is a marginal one. Hence, at $t \to -\infty$ we find the $SU(2)_k/U(1) \times \mathbb{R}_Q$ CFT. This is in accordance with the Weyl anomaly coefficient (4) for the case at hand (26). Using (27) this equals to

$$\text{computed as } t \to -\infty : \qquad W = 3 - 3w = 3 - \frac{6}{k} - \frac{6h_i^2}{k} = c_{2d} + c_{\ell.d.}. \tag{29}$$

Finally, we comment that contrary to the Euclidean case the system (25) does not admit solutions reaching $\lambda \to 1^-$ as $t \to 0^-$.

## 3 Scale invariant deformation of $SL(2,\mathbb{R})_{-k}/SO(2)$

In [16] a scale, albeit not Weyl invariant, deformation of the $SL(2,\mathbb{R})_{-k}/SO(2)$ coset CFT was constructed. The corresponding action is given by

$$\begin{aligned} S = \frac{k}{\pi} \int \mathrm{d}^2\sigma \Big\{ &\partial_+\rho\,\partial_-\rho - \coth^2\rho\,\partial_+\tau\,\partial_-\tau \\ &+\lambda\,\mathrm{e}^{2\tau}\,(\partial_+\rho + \coth\rho\,\partial_+\tau)(\partial_-\rho + \coth\rho\,\partial_-\tau) \Big\}. \end{aligned} \tag{30}$$

This model is scale invariant since it has a vanishing $\beta$-function [16]. However, it is not Weyl invariant since the required diffeomorphism (field redefinition) cannot be expressed in terms of a dilaton [24]. Equivalently, the corresponding energy-momentum tensor has non-vanishing trace. Indeed, for a metric background and a flat worldsheet the trace of the energy-momentum tensor is given by [7]

$$T_{+-} = \frac{1}{2k}\beta_{g_{\mu\nu}}\partial_+X^\mu\partial_-X^\nu, \quad \beta_{g_{\mu\nu}} = R_{\mu\nu} + 2\nabla_\mu\partial_\nu\Phi. \tag{31}$$

In our case the dilaton scalar is

$$\Phi = -\ln\sinh\rho, \tag{32}$$

yielding for the $\sigma$-model at hand the expression

$$T_{+-} = \frac{\lambda}{k}\mathrm{e}^{2\tau}(\partial_+\rho + \coth\rho\,\partial_+\tau)(\partial_-\rho + \coth\rho\,\partial_-\tau). \tag{33}$$

Finally, we note that this model can be obtained by taking an appropriate limit [16] to the $\lambda$-deformed $\sigma$-model corresponding to the non-compact coset $SL(2,\mathbb{R})_{-k}/SO(2)$.

### 3.1 Restoring conformal invariance

We add the term $k/\pi\,\partial_+ t\,\partial_- t$ to the above Lagrangian density and we allow $\lambda$ to depend on $t$. We also add to the scalar $\Phi$ (32) an additional term $\Phi_0(t)$. By imposing the one-loop beta-function equations for the metric (2) we find the following system of differential equations for $\lambda(t)$ and $h(t) = \dot{\Phi}_0(t)$

$$\ddot{\lambda} = 2h\dot{\lambda} + 4\lambda, \qquad \dot{h} = 0, \tag{34}$$

and the dilaton beta-function (3) yields the constant

$$-\frac{2}{k}\left(1+h^2\right)=w\,.\tag{35}$$

The above system is easily solved by

$$
\begin{aligned}
h(t)&=h_i\,,\quad \Phi_0(t)=h_i t\,,\\
\lambda(t)&=c_+\mathrm{e}^{a_+ t}+c_-\mathrm{e}^{a_- t}\,,\qquad a_\pm=h_i\pm\sqrt{h_i^2+4}\,,
\end{aligned}\tag{36}
$$

where $c_\pm$ and $h_i$ are integration constants. Without loss of generality we take $h_i>0$. The model necessarily has weakly and strongly coupled regions. Setting $c_-=0$, the function $\lambda(t)$ varies from zero as $t\to-\infty$, where the model is at weak coupling $\mathrm{e}^{\Phi(t)}\ll 1$, towards infinity as $t\to\infty$ where the model is at strong coupling $\mathrm{e}^{\Phi(t)}\gg 1$.

Following the analysis of Subsection 2.1 the holomorphic and anti-holomorphic dimension of the solution $\lambda(t)$ can be read using (15), (16) and (18), where in the case at hand

$$X=t\,,\qquad a=a_\pm\,,\qquad s=1\,,\qquad \alpha'=\frac{1}{k}\,,\qquad Q=h_i\,,\tag{37}$$

hence the holomorphic and antiholomorphic dimensions $(\Delta,\bar\Delta)$ equal to $-1/k$. When these are added up they provide the central charge deficit $-2/k$. In that respect, note that the perturbation is driven by a bilinear of the non-compact parafermions [25] whose holomorphic and anti-holomorphic dimensions are equal to $1+1/k$ adding up to $2(1+1/k)$. Hence, the aforementioned central charge deficit makes the perturbation a marginal one. In addition, we can also compute the central charge through the Weyl anomaly coefficient (4) for the case at hand (35)

$$W=c_{2d}+c_{\ell.d.}\,,\qquad c_{2d}=2+\frac{6}{k}+\cdots\,,\qquad c_{\ell.d.}=1+\frac{6h_i^2}{k}\,,\tag{38}$$

where we have decomposed the expression in two contributions, namely that of the scale invariant model (30) (to leading order in $1/k$) and that of the linear dilaton CFT with background charge $h_i$.

## 4 $\lambda$-deformed $SU(2)_k$

Let us now consider the $\lambda$-deformed $G_k$, whose action was introduced in [10]

$$S_{k,\lambda}(\mathfrak{g})=S_k(\mathfrak{g})+\frac{k}{\pi}\int \mathrm{d}^2\sigma\,[(\mathbb{1}-\lambda D^T)^{-1}\lambda]_{ab}J_+^a J_-^b\,,\tag{39}$$

where the action of the WZW model at level $k$

$$S_k(\mathfrak{g})=-\frac{k}{2\pi}\int \mathrm{d}^2\sigma\,\mathrm{Tr}(\mathfrak{g}^{-1}\partial_+\mathfrak{g}\,\mathfrak{g}^{-1}\partial_-\mathfrak{g})+\frac{k}{12\pi}\int \mathrm{Tr}\,(\mathfrak{g}^{-1}\mathrm{d}\mathfrak{g})^3\,,\tag{40}$$

as well as the (anti-)chiral currents

$$J_+^a=-i\,\mathrm{Tr}(t_a\partial_+\mathfrak{g}\,\mathfrak{g}^{-1})\,,\quad J_-^a=-i\,\mathrm{Tr}(t_a\mathfrak{g}^{-1}\partial_-\mathfrak{g})\,,\quad D_{ab}=\mathrm{Tr}\left(t_a\mathfrak{g}t_b\mathfrak{g}^{-1}\right)\,,\tag{41}$$

with $[t_a,t_b]=if_{abc}t_c$, $\mathrm{Tr}(t_a t_b)=\delta_{ab}$. In additon, the scalar $\Phi$ is given [26]

$$\mathrm{e}^{-2\Phi}=\det(\mathbb{1}-\lambda D^T)\,,\tag{42}$$

where we have ignored a constant proportionality factor. Note that we have given an alternative expression for the dilaton and for the action compared to the original literature to cover cases in which the matrix $\lambda$ is not invertible as in the Subsection 4.2 below.

## 4.1 Isotropic deformation

Let us now specialize to the isotropic deformation $\lambda_{ab} = \lambda \delta_{ab}$ for the $SU(2)$ case

$$\mathfrak{g} = e^{i\alpha n_a \sigma_a}, \quad n_a = \{-\sin\beta\sin\gamma, \sin\beta\cos\gamma, \cos\beta\}, \tag{43}$$

where $\sigma_a$ are the Pauli matrices. Inserting the above into (39) we find [10], in the normalization of (1), the metric

$$\begin{aligned}
&\mathrm{d}s^2 = 2k\left(\frac{1+\lambda}{1-\lambda}\mathrm{d}\alpha^2 + \frac{1-\lambda^2}{\Delta(\alpha)}\sin^2\alpha\left(\mathrm{d}\beta^2 + \sin^2\beta\,\mathrm{d}\gamma^2\right)\right), \\
&\Delta(\alpha) = (1-\lambda)^2\cos^2\alpha + (1+\lambda)^2\sin^2\alpha,
\end{aligned} \tag{44}$$

and the antisymmetric tensor

$$B = 2k\left(-\alpha + \frac{(1-\lambda)^2}{\Delta(\alpha)}\sin\alpha\cos\alpha\right)\sin\beta\,\mathrm{d}\beta \wedge \mathrm{d}\gamma. \tag{45}$$

The scalar $\Phi$ (42) equals to

$$\Phi = -\frac{1}{2}\ln\Delta(\alpha). \tag{46}$$

The above $\sigma$-model is renormalizable at one-loop in the $1/k$ expansion and its RG flow is given by [20, 21]

$$\frac{\mathrm{d}\lambda}{\mathrm{d}\ln\mu^2} = -\frac{c_G\lambda^2}{2k(1+\lambda)^2}. \tag{47}$$

where $c_G$ is the quadratic Casimir of the group. In the case at hand for $SU(2)$ the properly normalized generators are $t_a = \sigma_a/\sqrt{2}$ and $c_G = 4$. The latter RG flow describes an interpolation between $\lambda = 0$ in the UV towards the IR as $\lambda \to 1^-$ [20].

As before and concentrating on the Euclidean case, we add the term $k/\pi\,\partial_+ t\,\partial_- t$ to the corresponding Lagrangian density (1), we let the constant $\lambda$ to depend on $t$ and additionally we add to the scalar $\Phi$ (46) the term $\Phi_0(t)$. The one-loop equations for the metric and the antisymmetric tensor give the following system equations for $\lambda(t)$ and $h(t) = \dot{\Phi}_0(t)$

$$\begin{aligned}
&\ddot{\lambda} = -\frac{8\lambda^2}{(1+\lambda)^2} + \dot{\lambda}\left(2h - \frac{\lambda}{1-\lambda^2}\right), \\
&\dot{h} = \left(h\dot{\lambda} - \frac{4\lambda^2}{(1+\lambda)^2}\right)\frac{1-2\lambda}{1-\lambda^2} + \lambda\dot{\lambda}^2\frac{2-\lambda}{(1-\lambda^2)^2}.
\end{aligned} \tag{48}$$

In addition, the dilaton beta-function (3) yields the constant

$$\frac{2}{k}\left(\dot{h} - h^2\right) + \frac{\dot{\lambda}^2}{k}\frac{1-2\lambda}{(1-\lambda^2)^2} + \frac{2}{k}\frac{1+2\lambda+4\lambda^2-6\lambda^3+\lambda^4}{(1-\lambda)(1+\lambda)^3} = w. \tag{49}$$

The system (48) has as a trivial solution $\lambda(t) = 0, h(t) = $ constant, corresponding to the $SU(2)_k \times \mathbb{R}_Q$ CFT. Near that point the approximate solution valid for $t \to -\infty$ is

$$\lambda(t) \simeq c\,e^{2h_i t}, \quad \Phi_0(t) \simeq h_i t, \tag{50}$$

where $c, h_i$ are integration constants and the correction are as before exponential. Hence the model is at the weak coupling regime, i.e. $e^{\Phi(t)} \ll 1$. Following the analysis of Subsection 2.1, the holomorphic and anti-holomorphic dimension of the solution $\lambda(t)$ can be found through (18) where

$$X = t, \quad a = 2h_i, \quad s = 1, \quad \alpha' = \frac{1}{k}, \quad Q = h_i, \tag{51}$$

yielding $\Delta = \bar{\Delta} = 0$. Hence, the deformation around $\lambda = 0$ is indeed a marginal one. Hence, at $t \to -\infty$ we find the $SU(2)_k$ CFT times a linear dilaton background at charge $h_i$. Finally, inserting the latter into the Weyl anomaly constant coefficient (4) for the case at hand (49) we find that

$$\text{computed as } t \to -\infty: \quad W = 4 - 3w = 3 - \frac{6}{k} + 1 + \frac{6h_i^2}{k} = c_{3d} + c_{\ell.d.}, \tag{52}$$

corresponding to the central charge of the CFT $SU(2)_k$

$$c_{3d} = \frac{3k}{k+2} = 3 - \frac{6}{k} + \mathcal{O}\left(\frac{1}{k^2}\right), \tag{53}$$

plus central charge the linear dilaton background at charge $h_i$ respectively

$$c_{\ell.d.} = 1 + \frac{6h_i^2}{k}. \tag{54}$$

As in the renormalization group flow (7), the parameter $\lambda$ interpolates between $\lambda = 0$ as $t \to -\infty$ towards $\lambda \to 1^-$. To show the latter we expand the system (9) and the dilaton beta-function (10) as $\lambda(t) = 1 - \alpha(t)$ with $\alpha(t) \ll 1$. We obtain that

$$\ddot{\alpha} \simeq 2(2 + \dot{\alpha}h) + \frac{\dot{\alpha}^2}{\alpha}, \qquad \dot{h} \simeq \frac{\dot{\alpha}^2}{4\alpha^2}, \tag{55}$$

and

$$\frac{\dot{\alpha}^2}{4\alpha^2} - 2h^2 + \frac{2}{\alpha} \simeq kw. \tag{56}$$

We can solve the system (55) approximately as $t \to 0^-$ by

$$\alpha \simeq 2t^2, \qquad \Phi_0 \simeq -\ln(-t), \tag{57}$$

which corresponds to the constant $w = 0$. Since $e^{\Phi(t)} \gg 1$, the model becomes strongly coupled. Since, the dilaton beta-function is independent of $t$ using (52) for $w = 0$ we find that $h_i = 1$.

## 4.2 Marginal deformation

Next we specialize to the deformation corresponding the diagonal matrix $\lambda_{ab} = \text{diag}(0, 0, \lambda_3)$ for the $SU(2)$ case. This deformation is a marginal one and the background is conformal. We still consider this case so that we recover some known results in the literature. Parameterize the group element as

$$\mathfrak{g} = e^{i/2(\phi+\theta)\sigma_3} e^{i(\pi/2-\omega)\sigma_2} e^{i/2(\phi-\theta)\sigma_3}, \tag{58}$$

and inserting the above into (39) we find in the normalization of (1) the metric and the anti-symmetric tensor

$$ds^2 = 2k\left(d\omega^2 + \frac{(1-\lambda_3)\cos^2\omega d\theta^2 + (1+\lambda_3)\sin^2\omega d\phi^2}{1+\lambda_3\cos 2\omega}\right),$$
$$B = k\frac{\lambda_3 + \cos 2\omega}{1+\lambda_3\cos 2\omega}d\theta \wedge d\phi, \tag{59}$$

whereas the scalar is

$$\Phi = -\frac{1}{2}\ln(1 + \lambda_3\cos 2\omega). \tag{60}$$

This is a string background at one-loop in the $1/k$-expansion at it can be checked using (2), corresponding to the $SU(2)_k \times U(1)/U(1)$ gauged WZW model [27,28]. In addition, the deformation parameter $\lambda_3$ can be turned-on through an $O(2,2)$ transformation [29] on the exact $SU(2)_k$ string background.

Next, we dynamically promote the aforementioned string background by including the term $k/\pi\, \partial_+ t\, \partial_- t$ to the corresponding Lagrangian density (1) and taking $\lambda_3$ to be a function of $t$, adding to the dilaton $\Phi$ (60) the term $\Phi_0(t)$ and demanding conformality at one-loop (2) [14, 15] (see also [30] for earlier considerations)

$$\ddot{\lambda}_3 = \dot{\lambda}_3\left(2h - \frac{\lambda_3\dot{\lambda}_3}{1-\lambda_3^2}\right), \quad \dot{h} = -\frac{\lambda_3\dot{\lambda}_3 h}{1-\lambda_3^2}, \tag{61}$$

where $h(t) = \dot{\Phi}_0(t)$ and the dilaton beta-function (3) yields the constant

$$-\frac{2h^2}{k} + \frac{2}{k} + \frac{\dot{\lambda}_3(\dot{\lambda}_3 - 4\lambda_3 h)}{2k(1-\lambda_3^2)} = w. \tag{62}$$

The latter system of differential equations can be easily solved in general and has as a trivial solution $\lambda_3(t) = \text{constant}$ and $\Phi_0(t) = Q\,t$, hence it corresponds to the exact string background $SU(2)_k \times U(1)/U(1) \times \mathbb{R}_Q$.

In its Lorentzian version $t \to it$, it was shown in [15] that the corresponding string background can be mapped the Nappi–Witten exact CFT $\frac{SU(2)_k \times SL(2,\mathbb{R})_{-k}}{U(1) \times U(1)}$ [17]. For completeness we present the mapping in our parameterization. In particular Eqs.(14)-(17) in [17] with $(\psi, s, \lambda, \rho; k_{\text{there}}, \alpha)$ to the dynamical model (1), (59), (60) with $(t, \omega, \theta, \phi; k, \lambda_3(t))$, where

$$\psi = t, \quad s = \frac{\pi}{2} - \omega, \quad \rho = \phi, \quad \lambda = \theta, \quad k_{\text{there}} = 2k, \tag{63}$$

where $\lambda_3(t)$ and the dilaton $\Phi_0(t)$ are given by

$$\lambda_3(t) = \frac{\sin\alpha + \cos 2\psi}{1 + \sin\alpha\cos 2\psi}, \quad \Phi_0(t) = -\frac{1}{2}\ln(1 + \sin\alpha\cos 2\psi). \tag{64}$$

As a consistency check we have verified that the equations of motion (61) (they remain unaltered in the Lorentzian version) are satisfied from the above solution $\lambda_3(t), \Phi_0(t)$ and that the dilaton beta-function, taking $t \to it$ in (62), vanishes

$$\frac{2h^2}{k} + \frac{2}{k} - \frac{\dot{\lambda}_3(\dot{\lambda}_3 - 4\lambda_3 h)}{2k(1-\lambda_3^2)} = 0, \tag{65}$$

which was expected since the dilaton beta-function plays the rôle of the central charge at one-loop in the $1/k$-expansion through (4).

## 5 The $\lambda$-deformed $G_{k_1} \times G_{k_2}$

One may wonder if the promotion of the deformation parameters to functions of time is consistent with conformal invariance for a general deformation matrix $\lambda_{ab}$. In fact it is conceivable that these matrices may obey certain necessary conditions for conformal invariance to be restored. Since we are dealing with second order equations it is rather hard to provide an answer for the $\lambda$-deformed action (39). However, we have managed to do so for the $\lambda$-deformed action [18] which provides, for particular choices of the deformation matrix, new classes of integrable models, closely related, but distinct to those in [18]. The action is given by

$$S = S_{k_1}(\mathfrak{g}_1) + S_{k_2}(\mathfrak{g}_2) + \frac{k}{\pi}\lambda_{ab}\int \mathrm{d}^2\sigma\, J_{1+}^a J_{2-}^b, \tag{66}$$

where $k = \sqrt{k_1 k_2}$, $\mathfrak{g}_i$ are group elements of a semi-simple Lie group $G$ and $S_{k_i}(\mathfrak{g}_i)$ are the corresponding WZW actions at level $k_i$ (40). In addition

$$J_{i+}^a = -i \operatorname{Tr}(t_a \partial_+ \mathfrak{g}_i \mathfrak{g}_i^{-1}), \quad J_{i-}^a = -i \operatorname{Tr}(t_a \mathfrak{g}_i^{-1} \partial_- \mathfrak{g}_i), \quad [t_a, t_b] = i f_{abc} t_c, \quad \operatorname{Tr}(t_a t_b) = \delta_{ab},$$

with $a = 1, \ldots, d_G$ and the scalar $\Phi$ is a constant. In the isotropic case $\lambda_{ab} = \lambda \delta_{ab}$, the above $\sigma$-model is renormalizable at one-loop in the $c_G/k$ expansion and its RG flow is given by [18]

$$\frac{d\lambda}{d \ln \mu^2} = -\frac{c_G}{2k} \frac{\lambda^2 (\lambda - \lambda_0)(\lambda - \lambda_0^{-1})}{(1 - \lambda^2)^2}, \quad \lambda_0 = \sqrt{\frac{k_1}{k_2}}. \tag{67}$$

The above system of RG flows has apparently three fixed points namely: $\lambda = (0, \lambda_0, \lambda_0^{-1})$. Assuming that $k_2 > k_1$ or $\lambda_0 < 1$ it was shown in [18] that the action (66) interpolates between the UV fixed point $G_{k_1} \times G_{k_2}$ at $\lambda = 0$ and the IR fixed point $G_{k_2-k_1} \times G_{k_1}$ at $\lambda = \lambda_0$. Expanding the above $\beta$-function around the aforementioned fixed points we can read through (8) the classical scaling dimension of the driving operator around each fixed point [31,32]

$$\text{UV:} \quad \Delta_{\mathcal{O}} = 2, \quad \text{IR:} \quad \Delta_{\mathcal{O}} = 2 + \frac{c_G}{k} \frac{\lambda_0}{1 - \lambda_0^2}. \tag{68}$$

We emphasize that the model is renormalizable for a generic $\lambda_{ab}$ at one-loop in the $1/k$ expansion and its RG flow is given by [33]

$$\frac{d\lambda_{ab}}{d \ln \mu^2} = \frac{1}{2k} \mathcal{N}_{ac}{}^d \mathcal{N}_{bd}{}^{(T)c}, \tag{69}$$

where we use the definitions

$$\mathcal{N}_{ab}{}^c \equiv \mathcal{N}_{ab}{}^c(\lambda, \lambda_0^{-1}) = \left(\lambda_{ad}\lambda_{be} f_{def} - \lambda_0^{-1} \lambda_{ef} f_{abe}\right) g^{fc}, \quad \mathcal{N}_{ab}{}^{(T)c} = \mathcal{N}_{ab}{}^c(\lambda^T, \lambda_0), \tag{70}$$

with $g^{ab} = g_{ab}^{-1}$ with $g = \mathbb{I} - \lambda^T \lambda$ and $\tilde{g} = \mathbb{I} - \lambda \lambda^T$. The above RG fixed points persist even for non-isotropic deformations with general $\lambda_{ab}$.

## 5.1 Isotropic and marginal deformation

After adding the term $\frac{k}{2\pi} \partial_+ t\, \partial_- t$ to the corresponding Lagrangian density (1) for $\lambda_{ab} = \lambda \delta_{ab}$, we let the constant $\lambda$ to depend on $t$. As before we add to the constant scalar $\Phi$ the term $\Phi_0(t)$.

### 5.1.1 Isotropic deformation

Then, by imposing the one-loop equations (2) after specializing to the $SU(2)$ case we find the equations for $\lambda(t)$ and $h(t) = \dot{\Phi}_0(t)$

$$\begin{aligned}
\ddot{\lambda} &= 2h\dot{\lambda} - \frac{4\lambda^2(\lambda - \lambda_0)(\lambda - \lambda_0^{-1})}{(1 - \lambda^2)^2} + \frac{\lambda \dot{\lambda}^2}{1 - \lambda^2}, \\
\dot{h} &= \frac{6\lambda^3(\lambda - \lambda_0)(\lambda - \lambda_0^{-1})}{(1 - \lambda^2)^3} - \frac{3\lambda^2 \dot{\lambda}^2}{(1 - \lambda^2)^2} - \frac{3\lambda \dot{\lambda} h}{1 - \lambda^2},
\end{aligned} \tag{71}$$

where the latter system, as well as the corresponding action, is invariant under $t \to -t$. In addition, (71) has two trivial solutions, namely $\lambda(t) = 0$ and $\lambda(t) = \lambda_0$ with $h(t) = $ constant corresponding to the CFTs $SU(2)_{k_1} \times SU(2)_{k_2} \times \mathbb{R}_Q$ and $SU(2)_{k_1} \times SU(2)_{k_2-k_1} \times \mathbb{R}_Q$, respectively. In addition, the dilaton beta-function (3) gives the constant

$$-\frac{4h^2}{k} + \frac{2(\lambda_0 + \lambda_0^{-1})(1 - 3\lambda^2) + 8\lambda^3}{k(1 - \lambda^2)^3} + \frac{3(1 - 3\lambda^2)\dot{\lambda}^2}{k(1 - \lambda^2)^2} - \frac{12\lambda \dot{\lambda} h}{k(1 - \lambda^2)} = w, \tag{72}$$

from which we compute the central charge using the Weyl anomaly coefficient (4).

To further analyze the system (71), for the $SU(2)$ case, we consider with no loss of generality that $k_2 > k_1$. Around $t \to -\infty$ it has an approximate solution

$$\lambda(t) = c\, e^{2h_i t}, \quad \Phi_0(t) = h_i t + \mathcal{O}\left(e^{4h_i t}\right), \tag{73}$$

where $c$ and $h_i > 0$ are integration constants and where we have integrated for the dilaton field. Hence, the model is at weak coupling regime, i.e. $e^{\Phi(t)} \ll 1$. Following the analysis of Subsection 2.1, the holomorphic and anti-holomorphic dimensions of the solution $\lambda(t)$ can be found through (18) with

$$X = t, \quad a = 2h_i, \quad s = 1, \quad \alpha' = \frac{2}{k}, \quad Q = h_i, \tag{74}$$

yielding $\Delta = \bar{\Delta} = 0$. Hence, the deformation around $\lambda = 0$ is indeed a marginal one. Hence, at $t \to -\infty$ we find the $SU(2)_{k_1} \times SU(2)_{k_2}$ CFT times a free scalar at background at charge $h_i$. This is consistent with the Weyl anomaly coefficient (4) for the case at hand (72). Since $W$ is time independent we may compute it at any moment, in particular in the remote past for $t \to -\infty$ using (73). This leads to

$$\text{computed as } t \to -\infty: \quad W = 7 - \frac{6}{k}\left(\lambda_0 + \lambda_0^{-1}\right) + \frac{12h_i^2}{k}. \tag{75}$$

This corresponds to the central charge of the $SU(2)_{k_1} \times SU(2)_{k_2}$ CFT

$$c_{6d}^{(i)} = \frac{3k_1}{k_1 + 2} + \frac{3k_2}{k_2 + 2} = 6 - \frac{6}{k_1} - \frac{6}{k_2} + \mathcal{O}\left(\frac{1}{k_{1,2}^2}\right), \tag{76}$$

plus the central charge of a free scalar $t$ at a background charge $h_i$

$$c_{\ell.d.}^{(i)} = 1 + 6s\alpha' Q^2 = 1 + \frac{12h_i^2}{k}. \tag{77}$$

Unlike the models we have considered in previous sections in this case we may demand that in the remote infinity the solution reaches the IR fixed point at $\lambda = \lambda_0$, i.e. $\lambda(t)\big|_{t\to+\infty} = \lambda_0$. Expanding (71) around $\lambda_0$ we find the approximate solution

$$\lambda(t) = \lambda_0 + \tilde{c}\, e^{a_- t}, \quad h(t) = h_f + \mathcal{O}\left(e^{a_- t}\right), \quad a_- = h_f - \sqrt{h_f^2 + \frac{4\lambda_0}{1 - \lambda_0^2}} < 0, \tag{78}$$

where $h_f, \tilde{c}$ are integration constants. Integrating for the dilaton we find

$$\lambda(t) = \lambda_0 + \tilde{c}\, e^{a_- t}, \quad \Phi_0(t) = h_f t + \mathcal{O}\left(e^{a_- t}\right), \tag{79}$$

where $h_f < 0$ for validity of the solution, i.e. weak coupling regime $e^{\Phi(t)} \ll 1$. As before, the holomorphic and anti-holomorphic dimension of the solution $\lambda(t)$ can be found through (18) where

$$X = t, \quad a = a_-, \quad s = 1, \quad \alpha' = \frac{2}{k}, \quad Q = h_f, \tag{80}$$

leading to $\Delta = \bar{\Delta} = -\frac{2\lambda_0}{k(1-\lambda_0^2)}$. When these are added up they precisely provide the central charge deficit and the perturbation is a marginal one. Hence, at $t \to +\infty$ we find the WZW for $SU(2)_{k_2 - k_1} \times SU(2)_{k_1}$ CFT times a linear dilaton background at charge $h_f$. Then the Weyl

anomaly coefficient (4) which for the case at hand is (72) can be computed in the remote future at $t \to +\infty$ using (78). This leads to

$$\text{computed as t} \to +\infty: \quad W = 7 - \frac{6}{k} \frac{1}{\lambda_0(1-\lambda_0^2)} + \frac{12h_f^2}{k}, \tag{81}$$

which corresponding to the central charge of the $SU(2)_{k_1} \times SU(2)_{k_2-k_1}$ CFT

$$c_{6d}^{(f)} = \frac{3k_1}{k_1+2} + \frac{3(k_2-k_1)}{k_2-k_1+2} = 6 - \frac{6}{k_1} - \frac{6}{k_2-k_1} + \mathcal{O}\left(\frac{1}{k_{1,2}^2}\right), \tag{82}$$

plus the central charge of a free scalar $t$ at a background charge $h_f$

$$c_{\ell.d.}^{(f)} = 1 + \frac{12h_f^2}{k}. \tag{83}$$

The background charges $h_i$ and $h_f$ are related as the Weyl anomaly coefficient is independent of $t$. Equating the two expressions (75) and (81) we find that

$$h_f^2 - h_i^2 = \frac{\lambda_0^3}{2(1-\lambda_0^2)} > 0. \tag{84}$$

In conclusion, the system (71) interpolates between $\lambda = 0$ and $\lambda = \lambda_0$ corresponding to the CFTs $SU(2)_{k_1} \times SU(2)_{k_2}$ and $SU(2)_{k_1} \times SU(2)_{k_2-k_1}$ times the free scalar $t$ at background charge $h_i$ and $h_f$ respectively – see Figure 1.

$$t \to -\infty: \quad SU(2)_{k_1} \times SU(2)_{k_2} \quad \times \quad \mathbb{R}_{h_i}$$
$$\downarrow$$
$$t \to +\infty: \quad SU(2)_{k_1} \times SU(2)_{k_2-k_1} \quad \times \quad \mathbb{R}_{h_f}$$

Figure 1: CFT interpolation associated to (71). Numerical investigations showed that $\lambda(t)$ is monotonic.

### 5.1.2 Marginal deformation

In this Subsection we perturb the $SU(2)_k$ WZW along the marginal deformation $\lambda_{ab} = \text{diag}(0,0,\lambda_3)$. We note that in the static case the parameter $\lambda_3$ can be absorbed by an appropriate $O(4,4)$ duality transformation on the exact string background $SU(2)_{k_1} \times SU(2)_{k_2}$ [33]. Dynamically promoting $\lambda_3$ to depend on $t$ by adding $\frac{k}{2\pi}\partial_+ t\,\partial_- t$ to the corresponding Lagrangian density (1) and to the constant dilaton $\Phi$ the term $\Phi_0(t)$. By imposing the corresponding one-loop equations (2) for $\lambda_3(t)$ and $h(t) = \dot{\Phi}_0(t)$, which are given by (61) and the dilaton beta-function (3) yields the constant

$$-\frac{4h^2}{k} + \frac{2}{k}(\lambda_0 + \lambda_0^{-1}) + \frac{\dot{\lambda}_3(\dot{\lambda}_3 - 4\lambda_3 h)}{k(1-\lambda_3^2)} = w. \tag{85}$$

The corresponding $\sigma$-model for $|\lambda_3| < 1$ possesses no singularities and has a Euclidean signature. The system (61) has as a trivial solution $\lambda_3(t) = \text{constant}$ and $\Phi_0(t) = Q t$, hence it corresponds to the exact CFT $SU(2)_{k_1} \times SU(2)_{k_2} \times \mathbb{R}_Q$. Non-trivial time-dependent solutions

can be easily found, in particular those exhibited in Subsection 4.2 (for the Lorentzian case), i.e. in eq. (64), whose central charge can be read through (85) and (4), yielding

$$W = 7 - \frac{6(1+\lambda_0)^2}{k\lambda_0} = 7 - 6\left(\frac{1}{k_1} + \frac{1}{k_2} + \frac{2}{\sqrt{k_1 k_2}}\right), \tag{86}$$

corresponding to a seven-dimensional Euclidean CFT. It will be interesting to see if this CFT can be identified as it was done for the four-dimensional example studied in Subsection 4.2.

## 5.2 Generic deformation

We would like to generalize the previous analysis for a general deformation matrix $\lambda_{ab}(t)$ as well as a time dependent dilaton background $\Phi_0(t)$. The aim of this section is to determine under which conditions the full time dependent model has conformal symmetry. We use the one-loop equations (2) of the considered $\sigma$-model. Following the analysis of the Appendix A.2 we find that $\lambda_{ab}(t)$ and $h = \dot{\Phi}_0(t)$ should obey the differential equations (A.23) and (A.24) which are restated here for the reader's convenience

$$\ddot{\lambda}_{ab} = \dot{\lambda}_{ab} \mathrm{Tr}(\dot{\lambda} g^{-1}\lambda^T) - 2(\dot{\lambda} g^{-1}\lambda^T\dot{\lambda})_{ab} + \mathcal{N}_{ac}{}^d \mathcal{N}_{bd}{}^{(T)c} + 2\dot{\lambda}_{ab}h, \tag{87}$$

where the $\mathcal{N}_{ab}{}^c$'s are defined in (70) and also

$$2\dot{h} + \mathrm{Tr}(\ddot{\lambda} g^{-1}\lambda^T) + \mathrm{Tr}(\dot{\lambda} g^{-1}\lambda^T\dot{\lambda} g^{-1}\lambda^T) = 0. \tag{88}$$

These are accompanied by constraints on $\lambda_{ab}$, given in Eq.(A.25) (also reproduced here)

$$f_{abc}(g^{-1}\lambda^T\dot{\lambda} g^{-1})_{bc} = 0, \quad f_{abc}(\tilde{g}^{-1}\lambda\dot{\lambda}^T\tilde{g}^{-1})_{bc} = 0. \tag{89}$$

While (87), (88) describe the dynamical evolution of $\lambda_{ab}$ and $h$, the conditions (89) make sure that the whole construction is consistent with the one-loop conformality. In addition, the dilaton $\beta$-function (3) is given by (A.31)

$$\begin{aligned} w = {} & \frac{\lambda_0}{6k}\left(I_{abc}I_{pqr}\tilde{g}^{ap}\tilde{g}^{bq}\tilde{g}^{cr} + 3\mathcal{N}_{ab}{}^c\mathcal{N}_{pq}{}^r\tilde{g}^{ap}\tilde{g}^{bq}g_{cr}^2 + c_G \mathrm{d}_G\right) - \frac{4h^2}{k} \\ & + \frac{1}{k}\mathrm{Tr}(\dot{\lambda} g^{-1}\dot{\lambda}^T\tilde{g}^{-1}) - \frac{1}{k}\left(\mathrm{Tr}(\dot{\lambda} g^{-1}\lambda^T)\right)^2 - \frac{1}{k}\mathcal{N}_{ac}{}^d\mathcal{N}_{bd}{}^{(T)c}(g^{-1}\lambda^T)_{ab} \\ & - \frac{4}{k}\mathrm{Tr}(\dot{\lambda} g^{-1}\lambda^T)h, \end{aligned} \tag{90}$$

where the $I_{abc}$'s are given by (A.28), also presented here for reader's convenience

$$I_{abc} = \lambda_0^{-1}f_{abd}\tilde{g}_{cd} + \mathcal{N}_{bc}{}^d(g^{-1}\lambda^T)_{da} + \mathcal{N}_{ca}{}^d(g^{-1}\lambda^T)_{db}. \tag{91}$$

We would like to comment on possible consistent truncations of the general matrix $\lambda_{ab}$, in which some of its elements are set to zero. Conceptually, this is similar to truncations of the RG equations (69) whose existence is non-trivial. On one hand a consistent truncation in the system of RG flow equations is not necessarily a consistent one for the second order system (87), due to the second term in its right hand side. On the other hand, in the system of RG flows (69) we can also incorporate generic diffeomorphisms $\zeta^a$'s for consistency. These slightly modify (69) to [34]

$$\frac{\mathrm{d}\lambda_{ab}}{\mathrm{d}\ln\mu^2} = \frac{1}{2k}\mathcal{N}_{ac}{}^d\left(\mathcal{N}_{bd}{}^{(T)c} + g_{bd}\zeta^c\right). \tag{92}$$

However, we do not have this freedom in the dynamical (conformal) case, since these field redefinitions are not consistent with (2) for our dilaton ansatz $\Phi_0(t)$.

# 6 The $\lambda$-deformed $SU(2)_{k_1} \times SU(2)_{k_2}/SU(2)_{k_1+k_2}$

In this Section we dynamically promote the deformation parameter in an integrable model based on a coset CFT more complicated than that in Section 2. Our starting point will be the $\lambda$-deformed $SU(2)_{k_1} \times SU(2)_{k_2}/SU(2)_{k_1+k_2}$ [26], whose integrability was shown in [19]. The action is given by (1), where the metric and the field $\Phi$ are given by

$$
\begin{aligned}
\mathrm{d}s^2 = \frac{2(k_1+k_2)}{(1-\lambda)\Lambda}\Big( &\Omega_{\alpha\alpha}\mathrm{d}\alpha^2 + \Omega_{\beta\beta}\mathrm{d}\beta^2 + \Omega_{\gamma\gamma}\mathrm{d}\gamma^2 \\
&+ 2\Omega_{\alpha\beta}\mathrm{d}\alpha\mathrm{d}\beta + 2\Omega_{\beta\gamma}\mathrm{d}\beta\mathrm{d}\gamma + 2\Omega_{\alpha\gamma}\mathrm{d}\alpha\mathrm{d}\gamma \Big),
\end{aligned}
\tag{93}
$$
$$
\mathrm{e}^{-2\Phi} = \Lambda, \quad \Lambda = (1-\alpha^2)(1-\beta^2) - \gamma^2,
$$

with

$$
\begin{aligned}
\Omega_{\alpha\alpha} &= (1+\lambda_0^{-2})^{-2}Z^{-1}\big(Z^2 - \big(Z^2 - (1-\lambda)^2(1+\lambda_0^2)^2\big)\beta^2\big), \\
\Omega_{\beta\beta} &= (1+\lambda_0^2)^{-2}Z^{-1}\big(Z^2 - \big(Z^2 - (1-\lambda)^2(1+\lambda_0^{-2})^2\big)\alpha^2\big), \\
\Omega_{\gamma\gamma} &= (1-\lambda)^2 Z^{-1}, \\
\Omega_{\alpha\beta} &= (1-\lambda)^2 Z^{-1}\alpha\beta + (\lambda_0+\lambda_0^{-1})^{-2}Z\gamma, \\
\Omega_{\beta\gamma} &= -\lambda_0^{-2}(1-\lambda)^2 Z^{-1}\alpha, \quad \Omega_{\alpha\gamma} = -\lambda_0^2(1-\lambda)^2 Z^{-1}\beta,
\end{aligned}
\tag{94}
$$

and

$$
\lambda_0 = \sqrt{\frac{k_1}{k_2}}, \quad Z = 8\lambda + (1-\lambda)(\lambda_0+\lambda_0^{-1})^2.
\tag{95*}
$$

The above $\sigma$-model is renormalizable at one-loop in the $1/k$ expansion and its RG flow is given by [19]

$$
\frac{\mathrm{d}\lambda}{\mathrm{d}\ln\mu^2} = -\frac{c_G\lambda(1-\lambda_1^{-1}\lambda)(1-\lambda_2^{-1}\lambda)(1-\lambda_3^{-1}\lambda)}{2(k_1+k_2)(1-\lambda_f^{-1}\lambda)^2},
\tag{95}
$$

where

$$
\lambda_1 = \frac{1}{s_2 - 3s_1}, \quad \lambda_2 = \frac{1}{s_1 - 3s_2}, \quad \lambda_3 = \frac{1}{(s_1-s_2)^2}, \quad \lambda_f = \frac{1}{1-8s_1s_2},
\tag{96}
$$

and

$$
s_1 = \frac{\lambda_0^2}{1+\lambda_0^2}, \quad s_2 = \frac{1}{1+\lambda_0^2}.
\tag{97}
$$

The above RG flow equation has apparently four fixed points at $\lambda(t) = (0, \lambda_1, \lambda_2, \lambda_3)$. Assuming that $k_1 > k_2$ or $\lambda_0 > 1$ it was shown in [19] that the action (93) interpolates between the UV fixed point $G_{k_1} \times G_{k_2}/G_{k_1+k_2}$ at $\lambda = 0$ and the IR fixed point $G_{k_1-k_2} \times G_{k_2}/G_{k_1}$ at $\lambda = \lambda_1$. Expanding the above $\beta$-function around the aforementioned fixed points we can read through (8) the classical scaling dimension of the driving operator around each fixed point

$$
\text{UV}: \quad \Delta_{\mathcal{O}} = 2 - \frac{4}{k_1+k_2}, \quad \text{IR}: \quad \Delta_{\mathcal{O}} = 2 + \frac{4}{k_1-k_2}.
\tag{98}
$$

## 6.1 Restoring conformal invariance

Adding $\frac{k_1+k_2}{\pi}\partial_+ t\, \partial_- t$ to the corresponding Lagrangian density (1), we let the constant $\lambda$ to depend on $t$ and also add to the dilaton the term $\Phi_0(t)$. Imposing the one-loop equations (2)

we find the equations for $\lambda(t)$ and $h(t) = \dot{\Phi}_0(t)$

$$\ddot{\lambda} = 2h\dot{\lambda} - \frac{8\lambda(1-\lambda_1^{-1}\lambda)(1-\lambda_2^{-1}\lambda)(1-\lambda_3^{-1}\lambda)}{(1-\lambda_f^{-1}\lambda)^2} - 2\dot{\lambda}^2 \frac{(s_1-s_2)^2 - \lambda_f^{-1}\lambda}{(1-\lambda)(1-\lambda_f^{-1}\lambda)},$$

$$\dot{h} = \frac{24s_1^2s_2^2\dot{\lambda}^2}{(1-\lambda)^2(1-\lambda_f^{-1}\lambda)^2}. \tag{99}$$

These admit trivial solutions $\lambda(t) = (0, \lambda_1, \lambda_2, \lambda_3)$ and $h(t) = $ constant. In addition, the dilaton beta-function (3) yields a constant albeit rather lengthy expression which we will not present here.

To further analyze the system (99) we consider $k_1 > k_2$ or $\lambda_0 > 1$. Around $t \to -\infty$ the system is solved approximately by

$$\lambda(t) = c\,e^{a_-t}, \quad h(t) = h_i + \mathcal{O}\left(e^{2a_-t}\right), \quad a_- = h_i - \sqrt{h_i^2 - 8} > 0, \quad h_i > 2\sqrt{2}, \tag{100}$$

and integrating with respect to the dilaton we find that

$$\lambda(t) = c\,e^{a_-t}, \quad \Phi_0(t) = h_i t + \mathcal{O}\left(e^{2a_-t}\right). \tag{101}$$

showing that we are in the weak coupling regime, i.e. $e^{\Phi(t)} \ll 1$. The holomorphic and anti-holomorphic dimension of the solution $\lambda(t)$ can be found through (18) where

$$X = t, \quad a = a_-, \quad s = 1, \quad \alpha' = \frac{1}{k_1+k_2}, \quad Q = h_i, \tag{102}$$

leading to $\Delta = \bar{\Delta} = \frac{2}{k_1+k_2}$. Adding them up they precisely provide the central charge deficit and the perturbation is a marginal one. Hence, at $t \to -\infty$ we find the coset CFT $SU(2)_{k_1} \times SU(2)_{k_2}/SU(2)_{k_1+k_2}$ times the free scalar $t$ at background charge $h_i$. The Weyl anomaly constant coefficient computed as $t \to -\infty$ gives

$$\text{computed as } t \to -\infty: \quad W = 4 - \frac{6}{k_1} - \frac{6}{k_2} + \frac{6}{k_1+k_2} + \frac{6h_i^2}{k_1+k_2}, \tag{103}$$

corresponding to the central charge of the coset CFT $SU(2)_{k_1} \times SU(2)_{k_2}/SU(2)_{k_1+k_2}$

$$c_{3d}^{(i)} = \frac{3k_1}{k_1+2} + \frac{3k_2}{k_2+2} - \frac{3(k_1+k_2)}{k_1+k_2+2} = 3 - \frac{6}{k_1} - \frac{6}{k_2} + \frac{6}{k_1+k_2} + \mathcal{O}\left(\frac{1}{k_{1,2}^2}\right), \tag{104}$$

and the central charge of a free scalar $t$ at a background charge $h_i$

$$c_{\ell.d.}^{(i)} = 1 + \frac{6h_i^2}{k_1+k_2}. \tag{105}$$

Around $t \to +\infty$ the system is solved approximately by

$$\lambda(t) = \lambda_1 + \tilde{c}\,e^{\tilde{a}_-t}, \quad \Phi_0(t) = h_f t + \mathcal{O}\left(e^{2\tilde{a}_-t}\right),$$

$$\tilde{a}_- = h_f - \sqrt{h_f^2 + 8\frac{\lambda_0^2+1}{\lambda_0^2-1}} < 0, \tag{106}$$

where we have integrated with respect to the dilaton and where $h_f < 0$ for validity of the solution, i.e. weak coupling regime $e^{\Phi(t)} \ll 1$. Following the analysis of Subsection 2.1, the

holomorphic and anti-holomorphic dimension of the solution $\lambda(t)$ can be found through (18) where

$$X = t, \quad a = \tilde{a}_-, \quad s = 1, \quad \alpha' = \frac{1}{k_1 + k_2}, \quad Q = h_f, \tag{107}$$

leading to $\Delta = \bar{\Delta} = -\frac{2}{k_1 - k_2}$ whose sum precisely provides the central charge deficit for the perturbation to be marginal. Hence, at $t \to +\infty$ we find the coset CFT $SU(2)_{k_1-k_2} \times SU(2)_{k_2}/SU(2)_{k_1}$ times the free scalar $t$ at background charge $h_f$. The Weyl anomaly coefficient which as $t \to +\infty$ gives

$$\text{computed as t} \to +\infty: \quad W = 4 - \frac{6}{k_1 - k_2} - \frac{6}{k_2} + \frac{6}{k_1} + \frac{6h_f^2}{k_1 + k_2}, \tag{108}$$

corresponding to the central charge of the coset CFT $SU(2)_{k_1-k_2} \times SU(2)_{k_2}/SU(2)_{k_1}$

$$c_{3d}^{(f)} = \frac{3(k_1 - k_2)}{k_1 - k_2 + 2} + \frac{3k_2}{k_2 + 2} - \frac{3k_1}{k_1 + 2} = 3 - \frac{6}{k_1 - k_2} - \frac{6}{k_2} + \frac{6}{k_1} + \mathcal{O}\left(\frac{1}{k_{1,2}^2}\right), \tag{109}$$

and the central charge of a free scalar $t$ at a background charge $h_f$

$$c_{\ell.d.}^{(f)} = 1 + \frac{6h_f^2}{k_1 + k_2}. \tag{110}$$

In addition, the background charges $h_i$ and $h_f$ are related since the Weyl anomaly coefficient is independent of $t$. In particular, from (103) and (108) we find that

$$h_f^2 - h_i^2 = \frac{2}{\lambda_0^2(\lambda_0^2 - 1)} > 0. \tag{111}$$

In conclusion, the system (99) interpolates between $\lambda = 0$ and $\lambda = \lambda_1$ corresponding to the CFTs $SU(2)_{k_1} \times SU(2)_{k_2}/SU(2)_{k_1+k_2}$ and $SU(2)_{k_1-k_2} \times SU(2)_{k_2}/SU(2)_{k_1}$ times the free scalar $t$ at background charge $h_i$ and $h_f$ respectively – see Figure 2.

$$t \to -\infty: \quad SU(2)_{k_1} \times SU(2)_{k_2}/SU(2)_{k_1+k_2} \quad \times \quad \mathbb{R}_{h_i}$$
$$\downarrow$$
$$t \to +\infty: \quad SU(2)_{k_1-k_2} \times SU(2)_{k_2}/SU(2)_{k_1} \quad \times \quad \mathbb{R}_{h_f}$$

Figure 2: CFT interpolations associated to the system (71).

# 7 The $\eta$-deformed $SU(2)$ model

It is important to investigate whether or not the dynamical promotion of parameters may lead to conformal models if the starting points are integrable models albeit not the deformations of CFTs or coset CFTs, i.e. the $\lambda$-models. A natural playground to investigate this question are the $\eta$-deformed PCMs, in particular the $\eta$-deformed $SU(2)$ $\sigma$-model [12]

$$S = \frac{1}{2\pi T} \int d^2\sigma \left\{ \frac{R_+^1 R_-^1 + R_+^2 R_-^2}{1 + \eta^2} + R_+^3 R_-^3 \right\}, \tag{112}$$

where $R_\pm^a = -\frac{i}{\sqrt{2}} \text{Tr}(\sigma_a \partial_\pm g g^{-1})$ with $\sigma_a$'s the Pauli matrices, obeying the exterior algebra $dR^a = -\frac{1}{\sqrt{2}} \varepsilon_{abc} R^b \wedge R^c$. The above $\sigma$-model is renormalizable at one-loop order in the $T$ expansion and its RG flow is given by [35][3]

$$\frac{d\eta}{d\ln\mu^2} = T\eta(1+\eta^2)^2, \quad \frac{dT}{d\ln\mu^2} = -T^2(1+\eta^2)^2, \quad \eta\, T = \text{constant}. \tag{113}$$

## 7.1 Restoring conformal invariance

We add $\frac{1}{2\pi}\partial_+ t \partial_- t$ to the action (112) and let the parameters $\eta$ and $T$ to depend on $t$. In addition, we introduce the scalar $\Phi$ conveniently parameterized as

$$\Phi(t) = \Phi_0(t) - \frac{1}{4}\ln\left\{T^3(1+\eta)^2\right\}. \tag{114}$$

Demanding that the one-loop equations for conformal invariance (2) are satisfied, we find for the functions $\eta(t)$ and $h(t) = \dot{\Phi}_0(t)$ the system of differential equations

$$\begin{aligned}
\ddot{\eta} &= 2T\eta(1+\eta^2)^2 + \dot{\eta}\left(2h - \frac{1-\eta^2}{1+\eta^2}\frac{\dot{\eta}}{\eta}\right), \\
\ddot{T} &= -2T^2(1+\eta^2)^2 + \dot{T}\left(2h + \frac{\dot{T}}{T}\right), \\
\dot{h} &= \frac{3\dot{T}^2}{8T^2} + \frac{\eta\dot{\eta}}{1+\eta^2}\left(\frac{\dot{T}}{T} + \frac{\eta\dot{\eta}}{1+\eta^2}\right).
\end{aligned} \tag{115}$$

In addition, the dilaton $\beta$-function (3), yields the constant

$$-4h^2 + \frac{3\dot{T}^2}{4T^2} + T(3 + 2\eta^2 - \eta^4) + \frac{2\eta\dot{\eta}}{1+\eta^2}\left(\frac{\dot{T}}{T} + \frac{\eta\dot{\eta}}{1+\eta^2}\right) = w. \tag{116}$$

Note that these equations admit a constant dilaton $\Phi$ (114) as a solution which would correspond to a Ricci-flat background. Indeed, in this case, the first two equations in (115) are consistent and the third one yields the constraint (116) with $w = 0$, corresponding to a vanishing Ricci scalar. In this case, the corresponding second order equations admit a first-order formulation as differential equations, namely

$$\dot{\eta} = -\frac{\sqrt{2T}}{\eta}(1+\eta^2)\left(c + \eta^2 - c\sqrt{1+\eta^2}\right), \qquad \dot{T} = -\sqrt{2}T^{3/2}\left(1 - 2c - \eta^2\right), \tag{117}$$

where the parameter $c$ takes the values zero and one. For these two values, the above system is nothing but the Lagrange and Darboux–Halphen system, respectively. In addition, the corresponding background solution, which we will not present, describes the Eguchi–Hanson and the Taub–NUT self-dual gravitational instantons [37,38] endowed with an $\mathbb{R} \ltimes SU(2)$ structure. Finally, a comment is in order concerning the relation of the $\eta$-deformed $SU(2)$ model studied in the case at hand and the $\lambda$-deformed one studied in Section 4.1. As it is known, these models are related up to a Poisson–Lie T-duality and analytic continuation [35,39–42], namely

$$\lambda = \frac{i-\eta}{i+\eta}, \qquad k = \frac{i}{4\eta T}. \tag{118}$$

This map is not consistent with the dynamical equations (48) and (115). Specifically, the product $\eta T$ is time dependent whereas $k$ is not. Therefore, the Poisson-Lie duality does not commute with the insertion of time-dependence.

---

[3]These RG flow equations were also worked out in [36], leading however to the wrong sign in the first of them and consequently to the wrong conclusion that the ratio $\eta/T$ is constant under the RG flow, instead of the product as stated above.

# 8 Conclusions

In this present paper we studied the result of a modification of some important classes integrable $\sigma$-models under which the various constant parameters become dynamical, i.e. functions of time. The aim was to satisfy the conditions for conformal invariance at one-loop order. This is a reasonable demand but whether or not it would be materialized was far from clear. As a concrete realization, we first considered the $\lambda$-deformed models interpolating between a (gauged) WZW and the non-Abelian T-dual of a PCM [10]. The dynamically obtained model satisfies a system of non-linear second-order ordinary differential equations, whose trivial solutions are the fixed points of the RG flow of the parental model. Specifically, we considered the two-dimensional $\lambda$-deformed coset CFT $SU(2)_k/U(1)$ [10], interpolating from the coset CFT $SU(2)_k/U(1)$ at $t \to -\infty$ times a linear dilaton background to a strong coupling regime in which $\lambda(t) \to 1^-$, occurring at a finite value for $t$.

We applied the above set-up to the scale-invariant deformation of the $SL(2,\mathbb{R})_{-k}/SO(2)$ coset CFT [16] and to the $\lambda$-deformed $SU(2)_k$ [10]. In the latter case and for a dynamically promoted marginal deformation we obtained the Nappi–Witten expanding universe $\frac{SU(2)_k \times SL(2,\mathbb{R})_{-k}}{U(1) \times U(1)}$ [17], see also [14,15] for earlier considerations. In addition, we also considered the dynamical promotion of the Yang–Baxter deformed $SU(2)$ PCM [12] to a conformal model. In this case and for a constant dilaton the system of non-linear second-order ordinary differential equations admits a first-order sub-sector corresponding to the Eguchi–Hanson and the Taub–NUT self-dual gravitational instantons [37,38] endowed with an $\mathbb{R} \ltimes SU(2)$ structure. We extended the above results in the $\lambda$-deformed $G_{k_1} \times G_{k_2}$ and $SU(2)_{k_1} \times SU(2)_{k_2}/SU(2)_{k_1+k_2}$, which interpolate between exact (coset) CFTs [18,19]. The corresponding dynamical extension corresponds to a conformal model and it satisfies ordinary non-linear differential equations, which are trivially solved by the fixed points of the RG flow of the initial model. In addition, by appropriately choosing initial conditions the time-dependent conformal model interpolates between the RG fixed points as the time varies from the far past to the far future.

Let us emphasize that in the class of the $\lambda$-deformed WZW and gauged-WZW examples this procedure generates marginal deformations, at least at one-loop order. Note however, that in the $\eta$-deformed $SU(2)$ model, which was studied in Section 7, the dynamical deformation restores the conformal invariance, at least at one-loop order, since the model is a deformation of a PCM and not of a CFT.

Two comments are in order concerning the models studied in the present work. Firstly, although most of the models under consideration are classically integrable (apart from the generic ones in Section 5.2) we do not expect their dynamical promotion to preserve classical integrability. In that respect, we note that achieving conformality required the introduction of the dilaton field. Nevertheless, it would be of some interest to investigate separately the classical integrability of the aforementioned dynamical models. Secondly, we focused for simplicity on models based on rank-1 groups but the results of the present work can be extended for generic semisimple groups.

An immediate question is whether conformal invariance persists beyond the one-loop order. The simplest examples to consider would be the $\lambda$-deformed $G_{k_1} \times G_{k_2}$ [18] and also the $\eta$-deformed $SU(2)$ PCM [12]. We have studied the corresponding dynamically promoted models at two-loop order using the results of [43–47]. We found that conformality persists. We will not present the corresponding systems of ordinary non-linear differential equation, obeyed by the dynamically promoted deformation parameters and the dilaton, as they are rather complicated while conceptually there is not something new in their form. More precisely, in the $\lambda$-deformed $G_{k_1} \times G_{k_2}$ the derived system of differential equations admits as trivial solutions the RG fixed points $\lambda = 0, \lambda_0$. An asymptotic analysis, similar to that of Section 5.1.1, around those fixed points, can be performed ensuring the existence of an interpolating solution. Monotonicity of

the solution needs to checked independently using numerical methods.

Recall that, the system of non-linear second-order ordinary differential equations admits a first-order sub-sector in the case of dynamically promoted $\eta$-deformed $SU(2)$ PCM. It would be worth studying whether there exist analogue first-order sub-sectors for the other dynamically promoted models considered in the present work. This will facilitate the search for exact solutions for all values of time.

Finally, it would be worth extending the present set-up to generic $\sigma$-model $(G_{\mu\nu}, B_{\mu\nu}, \Phi)$ by promoting the constant moduli to time-dependent ones and demanding one-loop conformality. For generic $\sigma$-models this is a rather formidable task but given our analysis in Section 5.2, (and in Appendix A.2) we would expect in the cases that this works several consistency constraints (like (89)) aside from the dynamical evolution of the deformation parameters (the analogues (87) and (88)).

# Acknowledgements

We would like to thank C. Bachas and V. Spanos for discussions. The research work of K. Sfetsos and K. Siampos was supported by the Hellenic Foundation for Research and Innovation (H.F.R.I.) under the "First Call for H.F.R.I. Research Projects to support Faculty members and Researchers and the procurement of high-cost research equipment grant" (MIS 1857, Project Number: 16519).

# A    RG flows and geometry

## A.1    Generalities

Let us consider a $\sigma$-model of the form (1) whose metric and the two-form can be expressed in the tangent space, such that $G_{MN} = e^A{}_M e^B{}_N G_{AB}$ and $B_{MN} = e^A{}_M e^B{}_N B_{AB}$. This tangent space is spanned by the vectors $e_A = e_A{}^M \frac{\partial}{\partial X^M}$ and their dual one-forms $e^A = e^A{}_M dX^M$ satisfy the torsion-free condition

$$\nabla e^A = d e^A + \omega^A{}_B \wedge e^B = 0, \tag{A.1}$$

where $\nabla$ is the covariant exterior derivative and $\omega^A{}_B$ is the spin-connection. Similarly, we introduce the action of the covariant exterior derivative on a mixed tensor $V^A{}_B$

$$\nabla V^A{}_B = d V^A{}_B + \omega^A{}_C \wedge V^C{}_B - \omega^C{}_B \wedge V^A{}_C. \tag{A.2}$$

From the above we define the Riemann two-form $\Omega_{AB}$ and the Riemann tensor $R_{AB|CD}$

$$\nabla^2 V^A{}_B = \Omega^A{}_C \wedge V^C{}_B - \Omega^C{}_B \wedge V^A{}_C,$$
$$\Omega^A{}_B = \frac{1}{2} R^A{}_{B|CD} e^C \wedge e^D = d\omega^A{}_B + \omega^A{}_C \wedge \omega^C{}_B, \quad \Omega_{AB} = -\Omega_{BA}. \tag{A.3}$$

To compute the spin-connection we use (A.1) and we also assume the metricity of the covariant exterior derivative

$$\nabla G_{AB} = 0 \quad \Longrightarrow \quad d G_{AB} = \omega_{AB} + \omega_{BA}, \tag{A.4}$$

where we emphasize that $\omega_{AB}$ is not anti-symmetric except if the tangent space metric $G_{AB}$ is constant. Combining the latter with (A.2) yields the practical expression for the frame

components of the spin-connection

$$\omega_{AB} = \omega_{AB|C}\, e^C\,, \quad \omega_{AB|C} = \omega^0_{AB|C} + G_{AD}\Gamma_{BC}{}^D\,,$$
$$\omega^0_{AB|C} = \frac{1}{2}\left(C_{ABC} + C_{BCA} - C_{CAB}\right)\,,$$
$$\Gamma_{BC}{}^A = \frac{1}{2}G^{AD}\left(\partial_B G_{DC} + \partial_C G_{DB} - \partial_D G_{BC}\right)\,,$$

(A.5)

with $\partial_A = e_A{}^M \partial_M$ and we have also introduced the structure coefficients

$$de^A = \frac{1}{2}C^A{}_{BC}\, e^B \wedge e^C\,, \quad C^A{}_{BC} := -C^A{}_{CB}\,.$$

(A.6)

The above spin-connection can be generalized with the inclusion of a torsion term

$$\nabla^{\pm}e^A = \mp\frac{1}{2}T^A{}_{BC}\, e^B \wedge e^C\,, \quad \omega^{\pm A}{}_B = \omega^A{}_B \pm \frac{1}{2}T^A{}_{BC}\, e^C\,.$$

(A.7)

Using (A.6) and the torsion-full analogue of (A.3) we can compute the corresponding Riemann tensors

$$R^{\pm A}{}_{B|CD} = \partial_C \omega^{\pm A}{}_{B|D} - \partial_D \omega^{\pm A}{}_{B|C} + \omega^{\pm A}{}_{B|E}C^E{}_{CD} + \omega^{\pm A}{}_{E|C}\omega^{\pm E}{}_{B|D} - \omega^{\pm A}{}_{E|D}\omega^{\pm E}{}_{B|C}\,,$$

and the Ricci tensors

$$R^{\pm}_{AB} = \partial_C \omega^{\pm C}{}_{A|B} - \nabla^{\pm}_B \omega^{\pm C}{}_{A|C} - \omega^{\pm C}{}_{A|D}\omega^{\mp D}{}_{B|C}\,,$$

(A.8)

where $\omega^{\pm C}{}_{A|C}$ is a vector.[4] We may now rewrite the one-loop equations (2) in the compact form

$$R^-_{MN} + 2\nabla^+_N \partial_M \Phi = 0\,,$$

(A.9)

where the torsion-full Ricci tensor and the covariant derivative are built in terms of the torsion-full connections

$$\Gamma^{\pm}_{KL}{}^M = \Gamma_{KL}{}^M \pm \frac{1}{2}H^M{}_{KL}\,,$$

(A.10)

yielding

$$R^{\pm}_{MN} = R_{MN} - \frac{1}{4}H^2_{MN} \pm \frac{1}{2}\nabla^P H_{PMN}\,, \quad R^{\pm} = R - \frac{1}{4}H^2\,.$$

(A.11)

We can also rewrite (A.9) in the tangent space

$$R^-_{AB} + 2\nabla^-_B \partial_A \Phi = 0\,,$$

(A.12)

where

$$\nabla^-_B \partial_A \Phi = \partial_B \partial_A \Phi - \omega^{-C}{}_{A|B}\partial_C \Phi\,.$$

(A.13)

---

[4]To prove this we employ (A.7) and $\nabla_M e^M{}_A = 0$, yielding

$$\omega^{\pm C}{}_{A|C} = \partial_M e^M{}_A + \frac{1}{2}\partial_A \ln \det G_{MN} \pm \frac{1}{2}T^C{}_{AC}\,,$$

from which it can be easily shown that this transforms appropriately as a vector.

## A.2 Single $\lambda$-deformed $G_{k_1} \times G_{k_2}$

In this section we derive the differential equations that the time depended deformation matrix $\lambda_{ab}(t)$ and the dilaton background $\Phi = \Phi_0(t)$ should obey, for the construction to respect conformal invariance. Moreover, we obtain a number of constraints which ensure that the chosen background is consistent with conformality. We closely follow the analysis of [33]. The line element which represents the target space of the time depended model is given by[5]

$$\mathrm{d}s^2 = R^a R^a + \lambda_0^{-2} L^{\hat{a}} L^{\hat{a}} + 2\lambda_0^{-1} R^a L^{\hat{b}} + \mathrm{d}t^2 \,, \tag{A.14}$$

where

$$R^a = -i\,\mathrm{Tr}\left(t_a \mathrm{d}\mathfrak{g}_1 \mathfrak{g}_1^{-1}\right), \quad L^{\hat{a}} = -i\,\mathrm{Tr}\left(t_a \mathfrak{g}_2^{-1}\mathrm{d}\mathfrak{g}_2\right), \\ \mathrm{d}R^a = -\frac{1}{2}f_{abc}R^b \wedge R^c, \quad \mathrm{d}L^{\hat{a}} = \frac{1}{2}f_{abc}L^{\hat{b}} \wedge L^{\hat{c}}\,. \tag{A.15}$$

Hence, the unhatted and hatted indices denote the Maurer–Cartan forms of $\mathfrak{g}_1$ and $\mathfrak{g}_2$ respectively. By introducing the vielbeins

$$\mathrm{e}^a = R^a \,, \quad \mathrm{e}^{\hat{a}} = \lambda_{ba}(t)R^b + \lambda_0^{-1}L^{\hat{a}}\,, \quad \mathrm{e}^0 = \mathrm{d}t\,, \tag{A.16}$$

as well as the double index notation $A = (0, a, \hat{a})$ we move to the tangent space of the theory where the line element can be written as

$$\mathrm{d}s^2 = \tilde{g}_{ab}\mathrm{e}^a\mathrm{e}^b + \mathrm{e}^{\hat{a}}\mathrm{e}^{\hat{a}} + \mathrm{e}^0\mathrm{e}^0 = G_{AB}\mathrm{e}^A\mathrm{e}^B\,, \tag{A.17}$$

where $\tilde{g} = \mathbb{I} - \lambda\lambda^T$ and we have also defined $g = \mathbb{I} - \lambda^T\lambda$ for later convenience. Using these data one computes the spin connection using equations (A.5) as in [33]. Moreover, one needs to consider the torsion contribution from the $B$-field. The $B$-field is given by the following expression

$$B = B_0 + \lambda_0^{-1}\lambda_{ab}(t)R^a \wedge L^{\hat{b}}\,. \tag{A.18}$$

Hence, one finds for the torsion

$$H = -\frac{1}{6}(f_{abc} - 3f_{abd}(\lambda\lambda^T)_{cd} + 2\lambda_0\lambda_{ad}\lambda_{be}\lambda_{cf}f_{def})\mathrm{e}^a \wedge \mathrm{e}^b \wedge \mathrm{e}^c + \\ + \frac{1}{2}(\lambda_0\lambda_{bd}\lambda_{ce}f_{ade} - \lambda_{da}f_{dbc})\mathrm{e}^{\hat{a}} \wedge \mathrm{e}^b \wedge \mathrm{e}^c - \frac{\lambda_0}{6}f_{abc}\mathrm{e}^{\hat{a}} \wedge \mathrm{e}^{\hat{b}} \wedge \mathrm{e}^{\hat{c}} + \\ + \dot{\lambda}_{ab}\mathrm{e}^0 \wedge \mathrm{e}^a \wedge \mathrm{e}^{\hat{b}} - (\dot{\lambda}\lambda^T)_{ab}\mathrm{e}^0 \wedge \mathrm{e}^a \wedge \mathrm{e}^b\,, \tag{A.19}$$

where as in [33] we have

$$H_0 = \mathrm{d}B_0 = -\frac{1}{6}f_{abc}R^a \wedge R^b \wedge R^c - \frac{\lambda_0^{-2}}{6}f_{abc}L^{\hat{a}} \wedge L^{\hat{b}} \wedge L^{\hat{c}}\,. \tag{A.20}$$

Then from (A.7) we compute the new relevant components of the spin connection where we also used (A.4). Note that the components of the time independent model can be found in [33] and are still relevant for this analysis. The full result is

$$\omega_{ab}^+ = \left(-f_{abc} - \lambda_0\lambda_{ad}\lambda_{be}\lambda_{cf}f_{def} + (\lambda\lambda^T)_{ad}f_{dbc} + (\lambda\lambda^T)_{bd}f_{adc}\right)\mathrm{e}^c - (\dot{\lambda}\lambda^T)_{ab}\mathrm{e}^0\,, \\ \omega_{\hat{a}b}^+ = (\lambda_0\lambda_{bd}\lambda_{ce}f_{ade} - \lambda_{da}f_{dbc})\mathrm{e}^c - \dot{\lambda}_{ba}\mathrm{e}^0\,, \\ \omega_{\hat{a}\hat{b}}^+ = -\lambda_0 f_{abd}\lambda_{cd}\mathrm{e}^c\,, \\ \omega_{0a}^+ = (\lambda\dot{\lambda}^T)_{ab}\mathrm{e}^b\,,$$

---

[5]For simplicity, we have ignored an overall $k_1$ factor and also redefined $t \to \lambda_0^{1/2}t$.

$$\omega_{\hat{a}0}^+ = \dot{\lambda}_{ba}e^b\,, \tag{A.21}$$

$$\omega_{ab}^- = \left(\lambda_0\lambda_{ad}\lambda_{be}\lambda_{cf}f_{def} - (\lambda\lambda^T)_{cd}f_{dab}\right)e^c + (\lambda_{dc}f_{dab} - \lambda_0\lambda_{ad}\lambda_{be}f_{dec})e^{\hat{c}} - (\lambda\dot{\lambda}^T)_{ab}e^0\,,$$

$$\omega_{\hat{a}b}^- = 0\,,$$

$$\omega_{\hat{a}\hat{b}}^- = -\lambda_0 f_{abd}\lambda_{cd}e^c + \lambda_0 f_{abc}e^{\hat{c}}\,,$$

$$\omega_{0a}^- = (\dot{\lambda}\lambda^T)_{ab}e^b - \dot{\lambda}_{ab}e^{\hat{b}}\,,$$

$$\omega_{\hat{a}0}^- = 0\,.$$

Inserting the results into (A.8) we find the following components for the Ricci tensor

$$R_{a\hat{b}}^- = -\ddot{\lambda}_{ab} + \dot{\lambda}_{ab}\mathrm{Tr}(\dot{\lambda}g^{-1}\lambda^T) - 2(\dot{\lambda}g^{-1}\lambda^T\dot{\lambda})_{ab} + \lambda_0\mathcal{N}_{ac}{}^d\mathcal{N}_{bd}{}^{(T)c}\,,$$

$$R_{ab}^- = -R_{a\hat{c}}^-\lambda_{cb}^T\,,$$

$$R_{00}^- = \mathrm{Tr}(\tilde{g}^{-1}\ddot{\lambda}\lambda^T) + \mathrm{Tr}(\dot{\lambda}\lambda^T\tilde{g}^{-1}\dot{\lambda}\lambda^T\tilde{g}^{-1})\,,$$

$$R_{a0}^- = -\lambda_0\lambda_{ab}f_{bcd}(g^{-1}\lambda^T\dot{\lambda}g^{-1})_{cd} - f_{abc}(\tilde{g}^{-1}\lambda\dot{\lambda}^T\tilde{g}^{-1})_{bc}\,,$$

$$R_{0b}^- = \lambda_0\lambda_{bc}f_{cde}(g^{-1}\lambda^T\dot{\lambda}g^{-1})_{de} - (\lambda\lambda^T)_{bc}f_{cde}(\tilde{g}^{-1}\lambda\dot{\lambda}^T\tilde{g}^{-1})_{de}\,, \tag{A.22}$$

$$R_{0\hat{b}}^- = \lambda_0 f_{bcd}(g^{-1}\lambda^T\dot{\lambda}g^{-1})_{cd} + (\tilde{g}\lambda\dot{\lambda}^T\tilde{g}^{-1})_{de}\lambda_{cb}f_{cde}\,,$$

$$R_{\hat{a}0}^- = 0\,,$$

$$R_{\hat{a}b}^- = 0\,,$$

$$R_{\hat{a}\hat{b}}^- = 0\,,$$

where we used the identity $\lambda^T\tilde{g}^{-1} = g^{-1}\lambda^T$ and the $\mathcal{N}$'s are defined in (70).

Considering a background time dependent dilaton $\Phi_0(t)$ we obtain from (A.12) and (A.21) nine components corresponding to various values of $A = (0, a, \hat{a})$ and $B = (0, b, \hat{b})$. The resulting expressions are summarized in three different cases:

1. The components $a\hat{b}, ab$ and $00$ leading to the equations (A.23) and (A.24).

2. The components $0b, 0\hat{b}$ and $a0$ leading to the two constraints (A.25).

3. The components $\hat{a}0, \hat{a}b$ and $\hat{a}\hat{b}$ which trivially vanish.

Let us now analyze separately the first two cases. In the first case, the components $a\hat{b}, ab$ and $00$ yield the following differential equations for $\lambda_{ab}(t)$ and $h(t) = \dot{\Phi}_0(t)$

$$\ddot{\lambda}_{ab} = \dot{\lambda}_{ab}\mathrm{Tr}(\dot{\lambda}g^{-1}\lambda^T) - 2(\dot{\lambda}g^{-1}\lambda^T\dot{\lambda})_{ab} + \mathcal{N}_{ac}{}^d\mathcal{N}_{bd}{}^{(T)c} + 2\dot{\lambda}_{ab}h\,, \tag{A.23}$$

and

$$2\dot{h} + \mathrm{Tr}(\ddot{\lambda}g^{-1}\lambda^T) + \mathrm{Tr}(\dot{\lambda}g^{-1}\lambda^T\dot{\lambda}g^{-1}\lambda^T) = 0\,, \tag{A.24}$$

where we also took into account the redefinition of $t$ described in footnote 5. The above differential equations dictate the dynamical evolution of the deformation matrix and the dilaton field in a way which ensures that the deformed theory is conformally invariant at one-loop.

In the second case the components $0b, 0\hat{b}$ and $a0$, for which the dilaton contribution vanishes, lead to the following compatibility constraints

$$f_{abc}(g^{-1}\lambda^T\dot{\lambda}g^{-1})_{bc} = 0\,, \quad f_{abc}(\tilde{g}^{-1}\lambda\dot{\lambda}^T\tilde{g}^{-1})_{bc} = 0\,, \tag{A.25}$$

which are mapped to each other upon $\lambda \leftrightarrow \lambda^T$. These ensure that the model we consider is consistent with (A.12). They are trivially satisfied when $\lambda$ is part of the integrable sector, i.e. $\lambda_{ab} = \lambda\delta_{ab}$, reproducing the results of the previous sections. It is important to note, that searching for actual solutions of the above dynamical equations, one obtains new constraints.

In particular, starting from a deformation matrix $\lambda_{ab}$, with a number of vanishing entries, (A.23) indicates that these should not be introduced by the time evolution. This is an additional constraint which appears in special choices of the initial deformed theory, where the deformation is not switched on in every direction. The above requirements ensure that such a set up admits dynamical solutions. We have explicitly worked out the equations (A.23), (A.24) and the constraints (A.25) in several examples, including the $SU(2), SU(3), SU(4), G_2$ and $Sp(4)$ for different choices of the deformation matrix $\lambda_{ab}$. The resulting expressions are quite complicated and we will not present them here.

Finally, we also work out the dilaton beta function (3), which can be rewritten upon using (A.11), (A.13) and the conformality equation (A.12)

$$w = \frac{1}{6}H^2 + 2\nabla^2\Phi - 4\left(\partial\Phi\right)^2. \tag{A.26}$$

Reinserting the overall $k_1$ and also taking into account of footnote 5 we find $\left(\partial\Phi\right)^2 = {}^{h^2}\!/{}_k$ and we also evaluate $H^2$ using (A.19)

$$
\begin{aligned}
H^2 = {} & \frac{\lambda_0}{k}\left(I_{abc}I_{pqr}\tilde{g}^{ap}\tilde{g}^{bq}\tilde{g}^{cr} + 3\mathcal{N}_{ab}{}^c\mathcal{N}_{pq}{}^r\tilde{g}^{ap}\tilde{g}^{bq}g_{cr}^2 + c_G\mathrm{d}_G\right) \\
& + \frac{6}{k}\left(\mathrm{Tr}(\dot{\lambda}g^{-1}\dot{\lambda}^T\tilde{g}^{-1}) - \mathrm{Tr}(\dot{\lambda}g^{-1}\lambda^T\dot{\lambda}g^{-1}\lambda^T)\right),
\end{aligned}
\tag{A.27}
$$

where we have defined

$$I_{abc} = \lambda_0^{-1}f_{abd}\tilde{g}_{cd} + \mathcal{N}_{bc}{}^d(g^{-1}\lambda^T)_{da} + \mathcal{N}_{ca}{}^d(g^{-1}\lambda^T)_{db}. \tag{A.28}$$

Similarly, we evaluate $\nabla^2\Phi$ using (A.13) and (A.21)

$$\nabla^2\Phi = \frac{1}{k}\left(\dot{h} - \mathrm{Tr}(\dot{\lambda}g^{-1}\lambda^T)h\right). \tag{A.29}$$

Putting altogether into (A.26) we find

$$
\begin{aligned}
w = {} & \frac{\lambda_0}{6k}\left(I_{abc}I_{pqr}\tilde{g}^{ap}\tilde{g}^{bq}\tilde{g}^{cr} + 3\mathcal{N}_{ab}{}^c\mathcal{N}_{pq}{}^r\tilde{g}^{ap}\tilde{g}^{bq}g_{cr}^2 + c_G\mathrm{d}_G\right) - \frac{4h^2}{k} \\
& + \frac{1}{k}\left(\mathrm{Tr}(\dot{\lambda}g^{-1}\dot{\lambda}^T\tilde{g}^{-1}) - \mathrm{Tr}(\dot{\lambda}g^{-1}\lambda^T\dot{\lambda}g^{-1}\lambda^T)\right) + \frac{2}{k}\left(\dot{h} - \mathrm{Tr}(\dot{\lambda}g^{-1}\lambda^T)h\right).
\end{aligned}
\tag{A.30}
$$

or equivalently with the help of (A.23) and (A.24)

$$
\begin{aligned}
w = {} & \frac{\lambda_0}{6k}\left(I_{abc}I_{pqr}\tilde{g}^{ap}\tilde{g}^{bq}\tilde{g}^{cr} + 3\mathcal{N}_{ab}{}^c\mathcal{N}_{pq}{}^r\tilde{g}^{ap}\tilde{g}^{bq}g_{cr}^2 + c_G\mathrm{d}_G\right) - \frac{4h^2}{k} \\
& + \frac{1}{k}\mathrm{Tr}(\dot{\lambda}g^{-1}\dot{\lambda}^T\tilde{g}^{-1}) - \frac{1}{k}\left(\mathrm{Tr}(\dot{\lambda}g^{-1}\lambda^T)\right)^2 - \frac{1}{k}\mathcal{N}_{ac}{}^d\mathcal{N}_{bd}{}^{(T)c}(g^{-1}\lambda^T)_{ab} \\
& - \frac{4}{k}\mathrm{Tr}(\dot{\lambda}g^{-1}\lambda^T)h.
\end{aligned}
\tag{A.31}
$$

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
