# Peer review of "Dynamically restoring conformal invariance in (integrable) $σ$-models"

_SciPost Physics, doi:SciPost Phys. 14, 043 (2023)_

## Round 1 · Referee Report · Anonymous (Referee 1) · 2022-10-11

Strengths

Please see below

Weaknesses

Please see below

Report

The authors have clarified all the points which I had raised in my first report. I find the explanation very interesting, in particular I had not fully appreciated some of the subtleties which the authors have now emphasised. I would like to thank the author for adding these clarifying comments.

Requested changes

none

---

## Round 1 · Referee Report · Anonymous (Referee 2) · 2022-10-13

Report

In my first report I requested that the authors add comments in their paper on 3 issues in their resubmission. Unfortunately their resubmission does not include comments on all of these issues.

Issue 1: "It is not addressed whether the resulting 1-loop conformal theories are integrable (e.g. after gauge fixing)."

The authors state that "this was not the point of the present work" and do not include any comment in the new submission. Even if these questions cannot be addressed in the present work, it would be good to at least mention that they are questions of interest.
(If this work has nothing to do with integrability then why not just start from random non-integrable $\sigma$-models? What is the relevance of integrability?)

Issue 2: "The paper focuses on examples related to rank-1 groups. Would it be possible to consider λ-models based on general groups or cosets?"

The authors point out in their response that one of their examples addresses the general group case (not just rank 1). I would ask that they mention the possible generalization of all of their results to general simple groups somewhere in their submission.

Issue 3: "The generality of this picture is not discussed. It would be good to consider adding a time direction and inserting time-dependence in a general metric to see in which way the time-dependence must be inserted for conformal invariance. In this way, it could be possible to prove, or give evidence for, the claim of this paper in greater generality."

This issue has not been addressed in any way in the revised submission. Note that I am talking about general $\sigma$-models, not just the generalization $\lambda$-deformation with constant matrix $\lambda_{ab}$.

Since these points haven't been addressed, I must again request a revision.

---

## Round 1 · Author Response

Concerning the points raised by the first referee:

  1. The RG equations do not solve the system of second-order ordinary differential equations. Yet the system-order ordinary differential equations upon demanding an appropriate behaviour may interpolate between the fixed points.

We have added the following related comment, in the beginning of page 3 (Introduction):

"Let us point out ... between the fixed points."

  1. In the class of λ-deformed WZW and gauged-WZW examples (mostly considered in this work) this procedure indeed generates marginal deformations, at least at one-loop. However, in the example studied in Section 5, that is the $\eta$-deformed $SU(2)$ the dynamical deformation restores the conformal invariance as the model is a deformation of a PCM and not of a CFT.

We have added the following related comment, in the beginning of page 28 (Conclusion):

"Let us emphasize ... is a deformation of a PCM and not of a CFT."

  1. Concerning, the extension to all-loops as already stated in the conclusion we have checked that in the λ-deformed $G_{k_1}\times G_{k_2}$ and in the η-deformed deformed $SU(2)$ PCM, the conditions for conformal invariance at two-loop are consistent and lead to a more complicated system of second-order ordinary differential equations for the deformation parameters, which can be numerically integrated. More precisely, in the $\lambda$-deformed $G_{k_1}\times G_{k_2}$ the system of differential equations still admits as trivial solutions the RG fixed points $\lambda=0,\lambda_0$ and a similar asymptotic analysis, to that of Section 5.1.1, around those points can be performed ensuring the existence of an interpolating solution. Monotonicity of this solution has to be checked numerically.

We have added the following related comment, in the middle of page 28 (Conclusion):

"More precisely, in the $\lambda$-deformed $G_{k_1}\times G_{k_2}$ the ... independently using numerical methods."

%%%%%%%%%%%%%

Concerning the points raised by the second referee:

  1. Most of the models under consideration are integrable and are promoted to conformal ones by dynamically promoting their coupling constants and adding the dilaton. Note that in Section 5.2 (and Appendix A.2) we have also considered the generic deformation matrix \lambda_{ab} in equation (5.1), which is not expected to be integrable. Still the dynamically promoted model will be conformal upon satisfying equations (5.22), (5.23) and (5.24). We also note that there integrable cases, like the λ-deformed Yang-Baxter model studied in reference [32] for which the constraints (5.24) are not satisfied.

Concerning integrability, this was not the point of the present work. We have not explicitly checked whether the dynamically promoted models are classicaly integrable.

  1. a) As already mentioned above, in Section 5.2 (and in Appendix A.2) we have considered the generic deformation matrix \lambda_{ab} in equation (5.1) for generic semisimple groups (not just rank-), leading to the equations (5.22), (5.23) and (5.24). In the isotropic case $\lambda_{ab}=\lambda\delta_{ab}$ these equations are consistent and there is a related comment after equation A.25:

"They are trivially satisfied when $\lambda$ is part of the integrable sector, i.e. $\lambda_{ab}=\lambda \delta_{ab}$, reproducing the results of the previous sections."

b) In addition, we also considered coset cases, see Sections 2 and 3 (cosets of SU(2)k and SL(2,R)) but also Section 6 where we consider the non-diagonal coset SU(2){k_1}\times SU(2), for k_1 neq k_2. }/SU(2)_{k_1+k_2

c) Although the η-model and the λ-model are related up to a Poisson-Lie duality and an analytic continuation this ceases to be in the presence of time dependence. In particular, under Poisson-Lie duality the mapping of (λ,k) to (η,T) reads (see for example Eq.(6.2) in 1506.05784 for the su(2) case)

\lambda=\frac{i-\eta}{i+\eta}, k=\frac{i}{4\eta T}

This map is not compatible with the dynamical equations (4.10) and (7.4), hence the Poisson-Lie duality does not commute with the insertion of time-dependence.

We have added a related paragraph at the end of Section 7:

"Finally, a comment is order ... insertion of time-dependence."

  1. In the present work we demonstrated that the dynamical restoration of the conformal invariance works for several integrable and non-integrable cases of different dimensionality (groups, symmetric and non-symmetric cosets). Coming now to the point of restoration of conformality of a generic string background (G,B,\Phi), this appears to be at the moment a rather formidable task but given our analysis in Section 5.2 (and in Appendix A.2) we would expect constraints (like (5.24)) aside from the dynamical evolution of the deformation parameters (like (5.22) and (5.23)).

We have added footnote 2 after equation (1.4) (page 2) illustrating the general idea.

---

## Round 2 · Author Response

Issue 1: The seed models were chosen as an illustration of the method due to their simplicity (been integrable was an additional feature). Note that the dynamically promoted models are not expected to be classically integrable since demanding conformality introduces an additional one-loop factor that is the dilaton, i.e. $\dot\Phi_0(t)=h(t)$.

We have added the following related comment in the conclusion (page 28):

"Two comments are in order concerning ... of the aforementioned dynamical models."

Issue 2: The models in Sections 5.2 (generic semi-simple group) and 6 (non-symmetric coset) address to cases beyond rank-1 and symmetric cosets.

We have added the following related comment in the conclusion (page 28):

"Secondly, although we focused for ... semisimple groups."

and a comment at the end of page 34 - after equation (A.25)

"We have explicitly worked out ... we will not present them here."

Issue 3: We have added a related comment in the restoration of conformality of a generic string background (G,B,\Phi) in the last paragraph of the conclusion (page 29):

"Finally, it would be worth extending ... parameters (the analogues~\eqref{eq7.1} and~\eqref{eq7.2})."

---

## Editorial Decision

published